# Retrieve-Verify-Retrieve: Comprehensive Document Retrieval via Training Retriever for Iterative Search

## Abstract

Comprehensively retrieving diverse documents is crucial to address many queries that admit a wide range of valid answers. We introduce retrieve-verify-retrieve (RVR), a multi-round retrieval framework for comprehensive answer coverage that adopts an iterative approach equipped with retrievers and a verifier. The framework operates iteratively: (1) output documents from a previous retrieval step are verified by an LLM, (2) verified high-quality documents are appended to the query to act as input into the following retrieval round, and (3) the subsequent retriever, which was trained to target answers not yet covered by previous rounds, retrieves additional documents. RVR works effectively even with off-the-shelf retrievers, with fine-tuned retrievers bringing additional gains. Our approach outperforms baselines, including agentic search approaches, by at least 10% relative improvement and 3% absolute gain in complete recall percentage.

## 1. Introduction

Retrieval is a key method to equip large language models (LLMs) with up-to-date, long-tail information. Despite the recent advances in retrieval systems, comprehensively recovering relevant documents remains challenging (Amouyal et al., 2023; Chen & Choi, 2025).

In this work, we introduce Retrieve-Verify-Retrieve (RVR), a framework that performs multiple rounds of retrieval, where each round conditions on previously retrieved documents verified by an LLM. Documents deemed relevant by the verifier are concatenated to the query and also added to the final output set. At the next iteration, the model receives the concatenation of the query and successful outputs from

*Figure 1.* Overview of our Retrieve-Verify-Retrieve framework. Each query $q$ aims to retrieve documents to get multiple answers $(y_1, y_2, y_3)$. The initial retriever takes a query and returns document sets, and the verifier examines each document, identifying two valid answers $y_1, y_2$. The subsequent retriever takes the query and documents containing identified answers as input, targeting to retrieve complementary answer $y_3$.

the previous retrieval search, aiming to search for remaining gold documents. Figure 1 visualizes our approach.

Unlike traditional retrievers that only take a query as input to retrieve documents, our retriever is trained to explicitly condition on previously retrieved, high-quality documents to predict missing diverse documents. This formulation encourages iterative coverage rather than a one-pass similarity calculation between original queries and documents, reducing redundancy and improving diversity in the retrieved set. Our approach transforms retrieval from a one-shot ranking task into an iterative reasoning process over previously seen information.

Our approach is trained and evaluated on an open-domain, multi-answer QA benchmark (Amouyal et al., 2023). We further evaluate on another multi-answer QA benchmark (Malaviya et al., 2023), showing that our trained model can generalize to unseen distribution. We also evaluate recent agentic search framework (Jin et al., 2025; Team et al., 2025) which enables flexible workflow between LLMs and

[1]Anonymous Institution, Anonymous City, Anonymous Region, Anonymous Country. Correspondence to: Anonymous Author <anon.email@domain.com>.

Preliminary work. Under review by the International Conference on Machine Learning (ICML). Do not distribute.

retrieval modules, specifically allowing iterative search over the corpus, as baselines. We show consistent improvements in retrieval performance in both datasets. Although using off-the-shelf retrievers iteratively also showed some gains, a consistent, larger gain came from fine-tuning retrievers with our augmented input setting. We further provide a detailed ablation study of various model components. For example, to understand the impact of verifier performance, we provide performance under both our oracle verifier and our naive verifier (which simply take top K outputs from the previous retrieval). The experimental results show oracle verifier can further significantly boost the performance, showing the headroom of the approach with better verifier models.

In summary, we present a general iterative retrieval framework that models retrieval as a multi-round reasoning process conditioned on previously retrieved documents. Despite rich studies on agentic search, most approaches focus on improving LLMs that generate queries instead of adapting retrievals for a newer inference framework. We show that fine-tuning retrieval models for new inference scenarios can be crucial for further performance gains. Iterative retrieval thus represents a step toward retrieval systems that more closely emulate human information-seeking behavior, progressively refining their understanding of a query through reflection, verification, and iteration. We will release the code and model publicly upon publication.

## 2. Related Work

**Comprehensive Retrieval**   Recent work (Jiang et al., 2025) shows that there are many queries posed to LLMs that expect a wide range of valid answers. We review prior retrieval datasets with queries that admit a list of answers. QAMPARI (Amouyal et al., 2023) and QUEST (Malaviya et al., 2023) expect a list of entity answers from Wikipedia; Katz et al. (2023) expects all documents mentioning a list of entities given an entity type. Chen & Choi (2025) expect the candidate document set to cover all valid perspectives given a subjective question.

In terms of method, Min et al. (2021) introduces an autoregressive reranker conditioned on previously selected passages to encourage diversity and cover multiple answers. Chen et al. (2025a) proposed to generate multiple query embeddings autoregressively to retrieve more comprehensive document sets. In contrast to both, we chose iterative framework, where the output from previous retrieval step is provided as an input to the next retrieval stage, similar to recent agentic approaches.

**Iterative & Agentic Retrieval**   Several works have explored performing multiple rounds of retrieval for complex question answering (Yang et al., 2018; Trivedi et al., 2022).

Qi et al. (2019) introduced iterative query reformulation, generating new retrieval queries from partially read passages. A line of work (Xiong et al., 2021; Trivedi et al., 2023; Fang et al., 2025) similarly retrieve evidence iteratively, interweaving with intermediate LLM reasoning steps. A more recent line of works on agentic search systems (Jin et al., 2025; Team et al., 2025; Shao et al., 2025) explored using retrievers as a tool, and an LLM agent alternates between reasoning and tool calling until it reaches the final answer. All these works used a single fixed, off-the-shelf retriever system. Each retrieval stage operates as an independent query relying on handcrafted query updates or external reasoning modules. In contrast, we propose to train a retriever to retrieve missing documents after observing prior retrieval outputs.

**Verifier Based Retrieval**   Recent work has incorporated verification into retrieval pipelines. Chain-of-Verification (CoVe) (Dhuliawala et al., 2024) has an LLM draft an answer, plan verification questions, retrieve supporting evidence, and revise, reducing hallucinations. Self-RAG (Asai et al., 2024) integrates retrieval, generation, and critique via self-reflection signals. In these approaches, verification operates *after* retrieval or generation, serving primarily as a filtering mechanism. In contrast, our framework integrates verification *within* the retrieval loop itself: verification determines which retrieved documents are retained and used to condition subsequent retrieval rounds.

## 3. Method

### 3.1. Problem Formulation

**Multi Answer Retrieval Task**   Given a query $q$ and a large corpus $\mathcal{C} = \{d_1, \ldots, d_N\}$, a retrieval model $f(q, \mathcal{C}, K)$ should identify a ranked subset of $K$ documents $D_{\text{out}} = \{d_1, \ldots, d_K\} \subset \mathcal{C}$ that contains all answers $Y = y_1, y_2, ..., y_M$ for the query $q$. A retriever should aim to maximize both *relevance* and *coverage*, ensuring that the retrieved set reflects the full range of relevant, high-quality information.

**Metrics**   We assume a test set, where each query $q$ is annotated with a set of $M$ distinct gold answer strings $Y = y_1, y_2, ..., y_M$. We judge that document $d_i$ covers answer $y_j$ if the substring occurs in the document. We report the following two metrics related to task performance:

- **MRecall@K**: a binary score that equals 1 if all answers or at least $K$ answers in the answer set $\{y_1...y_M\}$ are covered by $D_{\text{out}}$. This metric was introduced in prior work (Min et al., 2021) for questions that admit multiple valid answers.
- **Recall@K**: the fraction of answers $Y$ that are covered

**Algorithm 1** Inference procedure for Retrieve-Verify-Retrieve framework.

---

**Input:** A input query $q$, corpus $\mathcal{C}$, initial retriever $f_i$, subsequent retriever $f_r$, a verifier $g$.

**Hyperparameters:** verifier budget per turn $B$, max document context budget $M$, maximum number of turns $T$, final output set size $K$.

**Output:** A ranked set of documents $D_{\text{out}} = \{d_1, \ldots, d_K\} \subset \mathcal{C}$, sorted by relevance to $q$.

---

1: $D_{\text{out}} \leftarrow \emptyset$
2: $D_i \leftarrow f_i(q, \mathcal{C}, K)$
3: **for** $t \in \{1, \ldots, T\}$ **while** $|D_{\text{out}}| < K$ **do**
4:      $D_v \leftarrow \{d \in D_i : g(d, q) \wedge \text{rank}(d) \leq B\}$     $\triangleright$ verified subset
5:      $D_{\text{out}} \leftarrow D_{\text{out}} \cup D_v$
6:      $D_{\text{ctx}} \leftarrow \text{TopK}(M, D_v)$         $\triangleright$ first $M$ for context
7:      $q_r \leftarrow [q; \bigoplus_{d \in D_{\text{ctx}}} d]$    $\triangleright$ concatenate query with context
8:      $D_i \leftarrow f_r(q_r, \mathcal{C}, K)$       $\triangleright$ peform $t+1$-th retrieval
9: **end for**
10: $D_{\text{out}} \leftarrow D_{\text{out}} \cup D_i$          $\triangleright$ add remaining
11: **return** $\text{TopK}(K, D_{\text{out}})$

---

- $\{d \in D : P(d)\}$: elements in $D$ satisfying predicate $P$
- $\text{TopK}(k, D)$: first $k$ elements of ordered set $D$
- $\bigoplus_{d \in D} d$: concatenation over ordered set $D$
- $\text{rank}(d)$: position of document $d$ in the retrieval ranking

by at least one document in $D_{\text{out}}$. This is a less stringent metric compared to MRecall@K.

### 3.2. Iterative Retrieval Framework

Algorithm 1 describes our iterative retrieval system. Our framework uses three components: an initial retriever $f_i$ that uses only the query, an iterative retriever $f_r$ that conditions on both the query and previously retrieved documents, and a verifier model $g$. Our subsequent retriever $f_r$ is a version of task-aware retrieval with instruction (Asai et al., 2022) which is trained to retrieve documents closer to the query $q$ but distinct from documents in the input ($\bigoplus_{d \in D_{\text{ctx}}} d$). The initial and subsequent retrievers can be the same model or different retrieval models, and we evaluate both possibilities in the experiments.

**Initial Retriever**    The initial retriever produces a ranked list of $K$ documents ($D_i = f_i(q, \mathcal{C}, K)$, line 2).

**Verifier**    Retrieval is often noisy, and most retrieved documents are not useful to answer the query. Thus, we implement a verifier $g$ that examines retrieval outputs and produces a verified subset $D_V = \{d \in D_i : g(d, q) \wedge \text{rank}(d) \leq B\}$ containing documents deemed relevant by verifier $g$, where $B$ is the verifier budget (line 4).

The verified subset, $D_V$, will constitute the final output

(line 10) and will be used in forming a query for the next stage retrieval. For subsequent retrieval turns, we select the top $M$ documents from the verified set (line 6) to form the concatenated query $q_r = [q; \bigoplus_{d \in D_{\text{ctx}}} d]$.

**Subsequent Retriever**    Using this query, the subsequent retriever produces new output $D_i = f_r(q_r, \mathcal{C}, B)$ (line 8). Using augmented query $q_r$ allows $f_r$ to reason about which relevant documents remain unretrieved, promoting answer coverage across iterations.

**Output**    To form the final output set, we accumulate verified documents across all rounds and add any remaining documents from the final iteration. Set semantics remove duplicates $D_{\text{out}} = \text{TopK}(K, \left( \bigcup_{t=0}^{T-1} D_V^{(t)} \right) \cup D_i^T)$ where $D_V^{(t)}$ denotes the verified set at round $t$, and we take only the top $K - |D_{\text{out}}|$ documents from the final retrieval $D_i^{(T)}$ to ensure the total output size is $K$.

### 3.3. Training

In this section, we describe the training of two sets of retrievers $f_i$ and $f_r$. We assume a training data, where each instance contains a query $q$ paired with a set of gold documents $D^*$ from corpus $\mathcal{C}$. We do not train the verifier, and use off-the-shelf LLMs. Our novelty lies in training the retriever $f_r$ with the proposed iterative inference scenario. We thus generate training data (positive and negative target documents) from this inference scenario. Besides that, we use standard contrastive retriever learning objective (Izacard et al., 2022):

$$\mathcal{L}_\theta = -\log \frac{\exp(s(f_\theta(x), f_\theta(d^+))/\tau)}{\sum_{d \in D_{batch}} \exp(s(f_\theta(x), f_\theta(d))/\tau)}$$

where retriever $f_\theta$ encodes the input query $x$, positive document $d^+$, and in-batch documents $d$, $s$ is a similarity function between the two embeddings, and $\tau$ is a temperature hyperparameter. We describe the process of constructing training data $(x, d^+, D_{batch})$ below.

**Training Data for Initial Retriever** ($D_i$)    For each query $q$, we randomly sample $d^+$ from our ground truth training data $D^*$. We also randomly sample one negative document $d^-$ from the corpus $\mathcal{C}$. Additionally, we leverage in-batch negatives (Karpukhin et al., 2020) such that all the other documents from other training examples in the same batch serve as additional negatives. Thus, $D_{batch}$ denotes all the documents in the batch, including the positive document, the sampled negative, and the in-batch negatives. We use the input query $q$ as is as $x$, forming $(x, d^+, D_{batch})$.

**Training Data for Subsequent Retriever** ($D_r$)    As discussed in Section 3.2, $f_r$ also takes gold documents as input

in addition to query $q$, where the number of appended gold documents is controlled by hyperparameter $M$. We first uniformly sample an integer $m \in \{0, ..., \min(M, |D^*|)\}$, and then sample $m$ documents from our ground truth set $D^*$ to form context $D_{\text{ctx}}$. The input query $x$ will be $[q; D_{\text{ctx}}]$. The positive document $d^+$ is randomly selected from the $D^* \setminus D_{\text{ctx}}$. Same as the training data for initial retriever, we randomly sample one negative document $d^-$ from the corpus $\mathcal{C}$, and leverage in-batch negatives.

### 3.4. Implementation Details

**Retriever**   For all experiments, we initialize the retrievers with pre-trained, off-the-shelf dual encoder retrievers: Contriever-MSMARCO (Izacard et al., 2022), Qwen3-Embedding-0.6B (Zhang et al., 2025), and INF-Retriever-v1-1.5B (Junhan Yang, 2025).

We use instances from the training portion of the QAMPARI dataset. Every model is fine-tuned for $50,000$ steps. We use a learning rate of $1 \times 10^{-4}$, temperature $\tau = 0.05$, batch size 48 per GPU, gradient accumulation of 2 steps, warm-up for 1K steps, and the AdamW optimizer (Loshchilov & Hutter, 2019). We use in-batch negatives and sample one negative document from the corpus. All experiments are conducted on NVIDIA H200 and L40 GPUs.

**Verifier**   We simply prompt an LLM (Qwen (Team, 2025)) [1] to serve as a verifier. Given an input document $d$ and a query $q$, the verifier outputs a binary label whether $d$ is relevant to the query $q$ or not. Formally, $g(d, q) = \{0, 1\}$. We use vLLM (Kwon et al., 2023) for inference. See Appendix C.1 for the prompt, and we discuss verifier performance in Section 6.3.

## 4. Experimental Settings

### 4.1. Dataset

**QAMPARI**   consists of open-domain questions paired with multiple valid answer strings (one example instance is shown in Fig 1). On average, each query is annotated with a set of 16.58 gold documents and contains 14.43 unique valid answers. Unlike most other datasets such as HotpotQA (Yang et al., 2018) and BrowseCompPlus (Chen et al., 2025b) which assume one gold answer, this dataset allows studying multi-answer coverage. We use their original split (training, evaluation).

**QUEST**   consists of queries paired with multiple valid answers, where each answer corresponds to a relevant Wikipedia document. The queries in QUEST specify set operations such as intersection, union, and difference (e.g., "what are some Films about bats that are not Superhero

films" (difference), "films that are South Korean adventure comedies or Canadian fantasy comedies" (union)).

**Retrieval Corpus**   For both datasets, we follow Amouyal et al. (2023) and use a Wikipedia dump from 2021-08-01 consisting of 25.9 million passages, averaging 100 words.

See Appendix A.1 and A.2 for full data statistics.

### 4.2. Comparison Systems

**Base** $f_i$   We evaluate three pre-trained off-the-shelf retrieval models without any fine-tuning.

**FT** $(D_i)$   Each pre-trained retriever is further fine-tuned on the QAMPARI training dataset with standard contrastive objective. This corresponds to training data for initial retriever in Section 3.3.

**Agentic Retrievers**   We implement two strong open-source agentic baselines, Tongyi DeepResearch agent[2] (Team et al., 2025) and SearchR1[3] (Jin et al., 2025). Both approaches use fixed retriever and train a LLM that generates search query. The former is continued pre-trained on agentic trajectories and post-trained on synthetic QA pairs. The latter is trained using PPO (Schulman et al., 2017) on Natural Questions (Kwiatkowski et al., 2019) and HotpotQA to maximize answer accuracy.

We use their trained model as is and follow their original implementations to retrieve $k_t = 5$ documents for Tongyi and $k_s = 3$ documents for SearchR1, respectively. To obtain the final candidate document set, we combine the documents returned for each retriever call without duplicates. For each agent, we run until it outputs an answer or it has collected over $K = 100$ documents in total. If the agent has only collected $K_a$ documents ($K_a < K$) upon termination, we additionally retrieve a set of $(K - K_a)$ documents using the last query issued to the retriever and append them to the candidate document list.

The exact prompts we use are provided in Appendix C.2 and C.3, and example trajectory traces are provided in Appendix C.4 and C.5. We experiment them with two retriever configurations, base retriever and fine-tuned retriever for fair comparison.

**Retrieve-Verify-Retrieve**   We use the following hyperparameters for our approach: $T = 2$ retrieval rounds, verifier budget $B = 100$, context budget $M = 3$ for Contriever-MSMARCO and $M = 6$ for INF-Retriever and Qwen3-Embedding. We evaluate three RVR configurations, differ-

---

[1] Qwen/Qwen3-30B-A3B-Instruct-2507

[2] Alibaba-NLP/Tongyi-DeepResearch-30B-A3B
[3] PeterJinGo/SearchR1-nq_hotpotqa_train-qwen2.5-7b-em-ppo

*Table 1.* Main Experimental Results on QAMPARI test set (N=1000). We report MRecall@100 and Recall@100. Our proposed methods outperform both the base retrievers (Base) and retrievers fine-tuned in-domain (FT ($D_i$)). RVR approaches also achieve much higher performance than agentic baselines.

| | Contriever-MSMARCO | | Base Retriever Model Qwen3-Embedding-0.6B | | INF-Retriever-v1-1.5B | |
| --- | --- | --- | --- | --- | --- | --- |
| | MR | R | MR | R | MR | R |
| Base $f_i$ | 19.00 | 54.17 | 16.70 | 52.94 | 26.10 | 62.34 |
| FT ($D_i$) | 28.60 | 63.19 | 26.90 | 63.48 | 29.30 | 65.99 |
| Tongyi (w/ Base $f_i$) | 6.60 | 35.96 | 16.20 | 52.22 | 20.40 | 57.03 |
| Tongyi (w/ FT ($D_i$)) | 9.80 | 42.09 | 22.00 | 60.67 | 21.30 | 58.13 |
| SearchR1 (w/ Base $f_i$) | 7.40 | 36.58 | 17.60 | 53.39 | 21.40 | 57.72 |
| SearchR1 (w/ FT ($D_i$)) | 9.60 | 40.83 | 24.30 | 57.82 | 27.00 | 60.96 |
| **Ours: Retrieve-Verify-Retrieve** | | | | | | |
| FT ($D_i$) + FT ($D_i$) | 28.80 | 63.59 | 30.30 | 66.80 | 31.10 | 66.76 |
| FT ($D_i + D_r$) + FT ($D_i + D_r$) | **31.70** | 66.12 | 26.30 | 61.24 | 32.40 | 68.04 |
| FT ($D_i$) + FT ($D_r$) | 31.60 | **66.83** | **31.10** | **67.12** | **33.70** | **68.70** |

*Table 2.* Efficiency comparison. We report the number of retrieval calls (# Calls) per query, as well as the seconds taken for retrieval (Ret.), the seconds taken for verification (Ver.) and total (sum of Ret. and Ver.). RVR methods are significantly more efficient than agentic baselines but still achieve better performance.

| | Contriever-MSMARCO | | | | Qwen3-Embedding-0.6B | | | | INF-Retriever-v1-1.5B | | | |
| --- | --- | --- | --- | --- | --- | --- | --- | --- | --- | --- | --- | --- |
| | # Calls ↓ | Ret.↓ | Ver.↓ | Sum↓ | # Calls↓ | Ret.↓ | Ver.↓ | Sum↓ | # Calls ↓ | Ret.↓ | Ver.↓ | Sum ↓ |
| Base | **1** | **1.91** | - | **1.91** | **1** | **1.34** | - | **1.34** | **1** | **3.75** | - | **3.75** |
| Tongyi | 14.94 | 97.72 | 92.32 | 190.04 | 15.85 | 232.79 | 83.60 | 316.39 | 16.42 | 264.30 | 80.40 | 344.70 |
| SearchR1 | 2.92 | 19.43 | 2.41 | 21.84 | 2.96 | 22.95 | 2.42 | 25.38 | 2.93 | 39.86 | 2.54 | 42.40 |
| RVR (w/ T=2, B=1) | 2 | 3.80 | 0.07 | 3.87 | 2 | 5.05 | 0.06 | 5.11 | 2 | 7.55 | 0.07 | 7.62 |
| RVR (w/ T=2, B=50) | 2 | 3.80 | 0.53 | 4.33 | 2 | 5.05 | 0.52 | 5.57 | 2 | 7.55 | 0.53 | 8.08 |
| RVR (w/ T=2, B=100) | 2 | 3.80 | 0.98 | 4.78 | 2 | 5.05 | 0.97 | 6.02 | 2 | 7.55 | 0.97 | 8.52 |

*Table 3.* Memory requirement comparison (all in GB), for LLM used ↓ and retriever index ↓. We report the size of each retriever model next to its name in the first row.

| | Contr. (0.48) | | Qwen3 (1.50) | | INF (5.43) | |
| --- | --- | --- | --- | --- | --- | --- |
| | LLM | Index | LLM | Index | LLM | Index |
| Base | - | 73.98 | - | 98.63 | - | 147.96 |
| Tongyi | 61.06 | 74.30 | 61.06 | 99.35 | 61.06 | 139.01 |
| SearchR1 | 17.41 | 73.98 | 17.41 | 98.63 | 17.41 | 147.95 |
| RVR ($f_i == f_r$) | 56.93 | 73.98 | 56.93 | 98.63 | 56.93 | 147.96 |
| RVR ($f_i != f_r$) | 56.93 | 147.96 | 56.93 | 197.26 | 56.93 | 295.92 |

ing in what is used for initial retriever ($f_i$) and subsequent retriever ($f_r$) : (1) FT ($D_i$) + FT ($D_i$): uses the same fine-tuned initial retriever $f_i$ in both rounds; (2) FT ($D_i + D_r$) + FT ($D_i + D_r$): uses a single model trained on the union of two retrieval training data ($D_i$ and $D_r$); (3) Fine-tuned ($D_i$) + Fine-tuned ($D_r$) : uses fine-tuned $f_i$ in round 1 and fine-tuned subsequent retriever $f_r$ in round 2. This is more costly as it requires two retrieval indexes and two retrieval models.

## 5. Results

### 5.1. In-Domain Results

**Task Performance** Table 1 reports retrieval performance on the QAMPARI test set. Fine-tuning with in-domain data improves raw performance significantly, especially improving weaker base retrievers (Contriever, Qwen) to be on-par with stronger base model (INF-Retriever). In-domain retriever improves the agentic approaches but overall the agentic approaches underperform the baseline in all three base model, even when paired with fine-tuned retriever. This could be caused by distribution shift from their query LLM training data, which typically requires multi-hop reasoning but not comprehensive answer coverage.

Our approach outperforms all baselines, raising both MRecall and R for all three base retrieval setting. Even using the same fine-tuned retriever $D_i$ as the baseline, verification step can bring gains when paired with stronger base retrievers (Qwen3, INF). Training retriever to collect complementary output $D_r$ further improves result. Having separate models (Fine-tuned ($D_i$) + Fine-tuned ($D_r$)) shows consistently strong results.

**Efficiency**   We evaluate the efficiency of our iterative retrieval framework on the QAMPARI test set across two dimensions: time and memory. For time, we report seconds per query (s/q) for both retrieval and verification/query generation components, measured with NVIDIA H200 GPUs. Retrieval time includes query encoding and k-nearest-neighbor search over the document index. Verification time depends on the verifier budget $B$ (number of documents verified per query) for our system, and the agentic search query generation process for agentic model. For memory, we report GPU memory usage in gigabytes for three components: the LLM verifier, the retrieval model, and the retrieval index.

Table 2 reports the time efficiency across different retrieval models and configurations. The default single-pass outperforms others, while the agentic search is substantially slower. Two-round retrieval with verification incurs additional overhead that scales with the verifier budget and number of calls, making it 2-3 times slower than baseline.

Table 3 shows memory requirements, reporting both RVR and agentic search requires additional GPU memory compared to baseline. Having a separate model for initial retriever and subsequent retriever is more costly.

### 5.2. Generalization to Other Dataset

We evaluate our models on a new dataset, QUEST. We find that fine-tuned $f_i$ underperforms the baseline, potentially due to domain shift. Therefore, we use the base retriever for our RVR framework. We evaluate two settings, one using base retriever for both initial and subsequent retriever, and another using base retriever for initial and Fine-tuned ($D_r$) for subsequent retriever. This fine-tuning is done in QAMPARI dataset. Table 4 report results for $B$=100 and Table 12 shows results for $B$=50. Our RVR largely outperforms baseline, especially when using retriever trained to find complementary results (Fine-tuned ($D_r$)).

## 6. Analysis

### 6.1. Ablation Studies on Retrievers

Table 5 presents ablation study of retriever components. In row 1 and 2, we report using off-the-shelf base retriever instead of fine-tuned retriever for $f_i$. As long as we fine-tune the subsequent retriever, the performance is quite competitive, sometimes even outperforming setting with both initial and subsequent retriever fine-tuned.

### 6.2. The contribution of 1st turn vs. 2nd turn retrieval

We analyze the individual contributions of our first-stage (base) and second-stage (iterative) retrievers. Table 6 presents a breakdown across our three models with $B$=100.

*Table 4.* **Out-of-Domain Generalization.** Results on QUEST test set (N=1727). We report MRecall@100 (MR) and Recall@100 (R). We report performances of different base retrieval models, agentic baselines, and RVR methods. For RVR, we use a verifier budget of 100.

|  | Contriever | | Qwen3 | | INF | |
|---|---|---|---|---|---|---|
|  | MR | R | MR | R | MR | R |
| Base | 3.24 | 23.79 | 3.13 | 21.38 | 4.75 | 26.60 |
| FT ($D_i$) | 3.13 | 18.67 | 2.43 | 18.38 | 4.75 | 26.31 |
| Tongyi | 0.93 | 10.51 | 2.49 | 19.52 | 3.30 | 21.95 |
| SearchR1 | 0.81 | 9.54 | 3.19 | 20.83 | 3.53 | 23.01 |
| **Ours: Retrieve-Verify-Retrieve** | | | | | | |
| Base + Base | 3.42 | 24.85 | 3.30 | 22.67 | 4.81 | 27.21 |
| Base + FT ($D_r$) | **4.52** | **26.01** | **4.52** | **25.86** | **6.02** | **30.53** |

*Table 5.* Ablation study on various RVR configurations with QAMPARI test set. Each row represents a configuration, denoted in the format of $f_i + f_r$ (initial + subsequent retriever). **Base**: off-the-shelf pretrained retriever; **FT**($D_i$): fine-tuned on initial retrieval data; **FT**($D_r$): fine-tuned on subsequent retrieval data with document context; **FT**($D_i + D_r$): fine-tuned on both objectives jointly.

|  | Contriever | | Qwen3 | | INF | |
|---|---|---|---|---|---|---|
|  | MR | R | MR | R | MR | R |
| Base + Base | 21.30 | 58.10 | 20.20 | 58.12 | 28.50 | 65.29 |
| FT($D_i$) + FT($D_i$) | 28.80 | 63.59 | 30.30 | 66.80 | 31.10 | 66.76 |
| Base + FT ($D_r$) | 30.70 | **67.25** | 29.20 | 66.36 | **34.20** | **69.29** |
| FT ($D_i$) + FT ($D_r$) | 31.60 | 66.83 | **31.10** | **67.12** | 33.70 | 68.70 |
| FT($D_i + D_r$) + FT($D_i + D_r$) | **31.70** | 66.12 | 26.30 | 61.24 | 32.40 | 68.04 |

Table 10 shows results for $B$=50.

The first-stage retriever provides the initial set of relevant documents. Across all models, the first stage contributes approximately 50 gold documents, corresponding to 7 unique answers, per question on average, demonstrating that initial retriever already captures a substantial portion of relevant information. The second-stage iterative retriever, conditioned on verified documents from the first stage, contributes an additional 22-26 gold documents, up to 1 unique answer, per question. This consistent improvement demonstrates that conditioning on previously retrieved context enables the model to discover initially missed documents.

### 6.3. The Impact of Verifier Performance

**Intrinsic Verifier Evaluation**   The goal of verifier is to identify retrieved documents that are relevant to the original query. To evaluate the verifier performance, we generate the following data: for each query in QAMPARI test set, we use three fine-tuned retrieval models $f_i$ from different base retrievers to retrieve 100 documents. We annotate whether they contain gold answer or not using the gold label set. For

*Table 6.* Contribution analysis of first and second stage retrieval on QAMPARI test set. We report the average number of new gold documents retrieved and the average number of new answers covered per question in each stage, with verifier budget of 100.

| | # New Gold Docs. | | # New Uniq. Ans. | |
|---|---|---|---|---|
| | Turn 1 | Turn 2 | Turn 1 | Turn 2 |
| Contriever | | | | |
| FT $(D_i)$ + FT $(D_i)$ | 49.59 | 17.60 | 7.12 | 0.33 |
| FT $(D_i)$ + FT $(D_r)$ | 49.59 | **22.83** | 7.12 | **0.73** |
| Qwen3 | | | | |
| FT $(D_i)$ + FT $(D_i)$ | 50.94 | 22.35 | 7.17 | 0.66 |
| FT $(D_i)$ + FT $(D_r)$ | 50.94 | **23.16** | 7.17 | **0.77** |
| Inf | | | | |
| FT $(D_i)$ + FT $(D_i)$ | 55.32 | 22.73 | 7.50 | 0.48 |
| FT $(D_i)$ + FT $(D_r)$ | 55.32 | **26.12** | 7.50 | **0.72** |

*Table 7.* Average verifier performance on top-100 retrieved documents on QAMPARI test set. Qwen3-30B-A3B-Instruct-2507 performs the best in terms of recall and is used as our verifier for the main experiments.

| VERIFIER | PRECISION | RECALL | ACCURACY |
|---|---|---|---|
| RANDOM BASELINE | 6.71 | 6.42 | 64.85 |
| GPT-5-NANO | 50.25 | 54.70 | 82.29 |
| QWEN3-4B-INST | 43.62 | 52.57 | 77.63 |
| QWEN3-30B-INST | 34.06 | 74.05 | 68.43 |

this set, we report 21.07% positive and 78.93% negative documents.

We report the performance of three LLMs, GPT-5-nano, Qwen3-4B-Instruct-2507 (Team, 2025), and Qwen3-30B-A3B-Instruct-2507 (Team, 2025), as verifier on this dataset. Table 7 presents the results. Qwen3-30B-A3B-Instruct-2507 model achieves the highest recall, which aligns with our objective of maximizing retrieval coverage. Therefore, we adopt Qwen3-30B-A3B-Instruct-2507 as the verifier for main experiments.

**Extrinsic Verifier Evaluation**   In this section, we isolate the impact of verifier performance in end-to-end retrieval performance.

We evaluate verifier performance by comparing against a baseline and an upper bound. As a baseline (TopK), we use the top $M$ documents ranked by the initial retriever to form the query context without a verifier. The final output set (K=100) combines the top 50 documents from the initial retriever with the top 50 non-duplicate documents from the subsequent retriever.

As an upper bound, we use oracle verifier that has access to gold answer string labels. For each example, we have a set of answer strings. Our oracle verifier leverages these

*Table 8.* Comparing the performance of using oracle vs. LLM verifier (Qwen3-30B) on QAMPARI test set. We evaluate MRecall@100 in the RVR setting FT $(D_i)$ + FT $(D_r)$, with a verifier budget of 100. Using LLM (Qwen3) verifier comes close to using an oracle verifier (upper bound) and outperforms TOPK (baseline).

| VERIFIER | CONTRIEVER | QWEN3 | INF |
|---|---|---|---|
| ORACLE | **33.80** | **33.60** | **36.30** |
| LLM (QWEN3-30B) | 31.60 | 31.10 | 33.70 |
| TOPK | 26.90 | 27.50 | 28.50 |

answer strings to determine which documents are gold. If a document contains any of the answer strings, it is considered a gold document. This serves as an upper bound on verifier performance.

Table 8 presents the comparison across verifier settings: oracle, LLM (Qwen3-30B), and TopK. Across all base models, we see noticeable gains with oracle verifier, suggesting performance gain can be achieved by improving the verifier.

### 6.4. Multi-turn Generalization

Figure 2 compares performance across five iterations using both LLM and oracle verifiers. The left panel shows results with our LLM verifier (Qwen3-30B), where gains plateau after the second iteration. The right panel shows results with an oracle verifier that selects documents containing unique answer strings for $D_{ctx}$, demonstrating steady improvements across all five iterations. This disparity suggests the LLM verifier tends to select redundant documents in later iterations, even when the retrieved set contains documents with unique answers. These results indicate that improved verification mechanisms could substantially enhance RVR performance.

### 6.5. The Impact of Verifier Budget

Figure 3 plots the system performance for varying verifier budget (10, 20, 50, 100 documents) to evaluate models under more resource-constrained settings. The scores for 100 document setting is the same as reported in Table 1.

We compare two settings of our Retrieve-Verify-Retrieve, one with FT$(D_i)$ and another with FT$(D_i)$ and FT$(D_r)$. Across all models, the absolute performance decreases as the verifier budget shrinks. Using fine-tuned subsequent retriever FT$(D_r)$ was particularly helpful in more restrictive budget setting.

### 6.6. The Impact of Varying Input Length

We study the impact of varying the maximum number of input documents provided to the subsequent retriever. Here, we do not report on Contriever, as it has a sequence length limit of 512 tokens, resulting in truncation when we provide

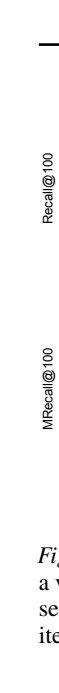
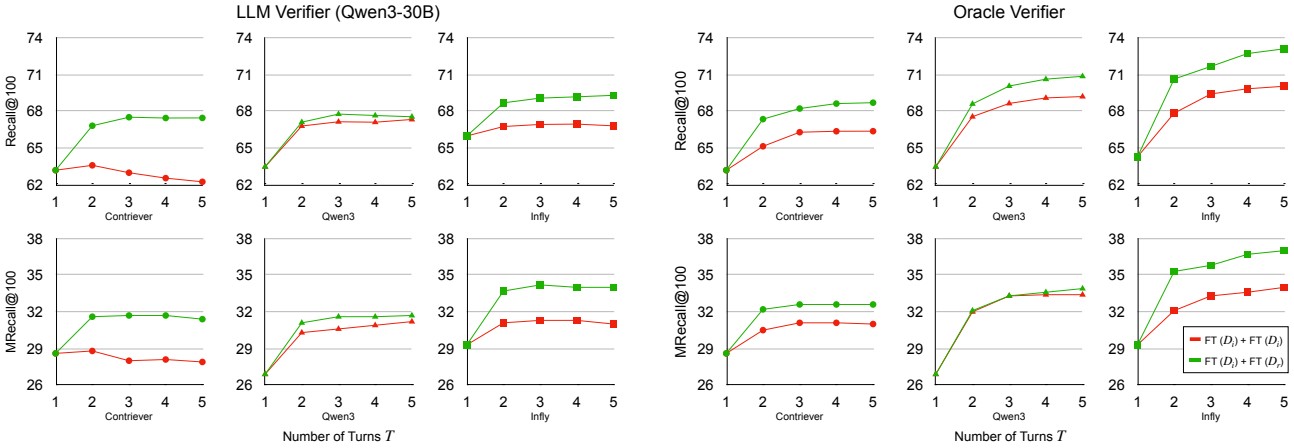

Figure 2. Multi-turn Generalization Results. This figure illustrates the change in Recall@100 and MRecall@100 across five iterations with a verifier budget of 100. Left panels show results with LLM verifier (Qwen3-30B), while right panels show results with oracle verifier that selects documents containing unique answer strings to be used as input. Performance with the LLM verifier plateaus after the second iteration, whereas the oracle verifier shows continued improvement, indicating substantial headroom for better verification mechanisms.

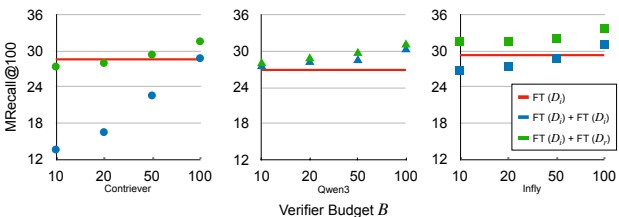

Figure 3. Varying the verifier budget. We evaluate MR@100 across three new verifier budgets on QAMPARI dataset. RVR is shown in green and blue, while our one-round baseline in red. See Appendix B.1 for Recall@100 results.

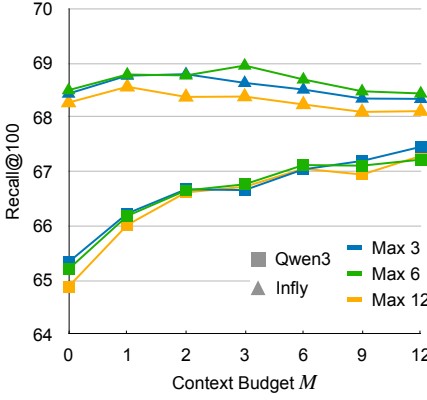

Figure 4. The performance (Recall@100) with varying number of input documents at inference time (Context Budget $M$). We compare models fine-tuned with different maximum document counts (3, 6, and 12 docs) for INF and Qwen3. Different colors denote fine-tuning with different number of documents, and shapes indicate the retrievers used. Other metric results (MRecall@100) are provided in Appendix B.2.

more than 3 input documents. We run inference with our RVR model (Fine-tuned + Ours) setting, fine-tuned with a maximum of 3, 6, and 12 input documents. We further vary the context budget, $M$, used during inference time.

Figure 4 displays the trends, as we sample from 0 up to 12 input documents. We find that increasing the number of documents at inference time beyond 6 does not show strong gains. We also observe that on average, models fine-tuned with up to 6 input documents perform better.

## 7. Conclusion

We introduced Retrieve-Verify-Retrieve framework, a framework that conditions on previously retrieved evidence and explicitly optimizes for answer coverage. By training retrievers to predict missing gold documents and integrating verifier-guided iteration, our approach consistently expands retrieval coverage while reducing redundancy. Results on QAMPARI and zero-shot generalization to QUEST demonstrate that iterative conditioning provides a robust and general mechanism for improving retrieval completeness.

## Impact Statement

This paper introduces a retrieval framework designed to improve answer coverage and reduce redundancy in information retrieval systems. By enabling retrieval models to iteratively reason over previously retrieved evidence, the proposed approach may improve the completeness and reliability of systems used in applications such as search engines, question answering, and knowledge discovery.

Potential positive impacts include more thorough information access, reduced bias toward dominant viewpoints, and

improved support for users seeking comprehensive answers. As with any retrieval technology, the quality of results ultimately depend on the underlying data sources and the design of downstream systems. Potential risks mirror those of existing retrieval systems, including surfacing misinformation or amplifying biases present in the corpus.

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

*Table 9.* Dataset statistics showing the number of questions split across train, development, and test sets, along with the average number of answers and gold documents per question.

| | # examples | | | # answers | # gold docs |
|---|---|---|---|---|---|
| **Dataset** | Train | Dev | Test | Avg. | Avg. |
| QAMPARI | 34,425 | 6,191 | 1,000 | 14.43 | 16.58 |
| QUEST | — | — | 1,727 | 10.50 | — |

## A. Datasets

### A.1. QAMPARI

Table 9 displays our dataset statistics in detail. An example of a gold document from our QAMPARI test set, along with the query and answer strings is below.

**Query**: What manga was drawn by Ryoichi Ikegami?

**Answer Strings**: Heat, *Mai, the Psychic Girl*, Wounded Man, Sanctuary, Crying Freeman, Strain

**Document Text**: *Mai, the Psychic Girl*, known simply as in Japan, is a manga written by Kazuya Kudo and illustrated by Ryoichi Ikegami. The main character is Mai Kuju, a 14-year-old Japanese girl with powerful psychic abilities. She is being pursued by the Wisdom Alliance, an organization which secretly strives to control the world. The alliance already controls four other powerful psychic children, and it has hired the Kaieda Intelligence Agency to capture Mai. Media. Manga. *Mai, the Psychic Girl* is one of the first manga series to be fully published in English.

### A.2. QUEST

Below, we show an example query from the QUEST test set, its associated answer string, and a document from our corpus that contains this answer. Table 9 shows additional dataset statistics.

**Query**: what are some 1950s comedy mystery or spy comedy films

**Answer Strings**: The Hole (1957 film), Charade (1953 film), Hot Stuff (1956 film), The Trouble with Harry, Father Brown (film), The Runaway Bus, Boston Quackie, *My Favorite Spy*, The Fuller Brush Girl, Scared Stiff (1953 film), Our Man in Havana (film), Spy Chasers, Mrs. O'Malley and Mr. Malone, Clipped Wings (1953 film), Commotion on the Ocean, Down Among the Z Men, Knock on Wood (film), Top Secret (1952 film)

**Document Text**: *My Favorite Spy* is a 1951 comedy film directed by Norman Z. McLeod and starring Bob Hope and Hedy Lamarr. Plot. US intelligence agents recruit burlesque comic Peanuts White to pose as international spy Eric Augustine, whom he resembles, to acquire a million-dollar microfilm in Tangier. There, he encounters the irresistible Lily Dalbray, Augustine's one-time friend, who is now in league with his arch-enemy, Brubaker.

### A.3. Instruction Used for Instruction Tuned Models

We use the same instruction to finetune our Qwen3-0.6B-Embed and inf-retriever-1.5B models: *Given a query, retrieve relevant passages that answer the query*. We use this instruction during both fine-tuning and inference.

## B. More Analysis

### B.1. Impact of Verifier Budget on Recall@K

Figure 5 displays Precision@100 and Recall@100 with verifier budgets of 10, 20, 50, and 100.

### B.2. Impact of Varying Input Documents on MRecall@K

Figure 6 displays additional results for our experiment where we vary the number of input documents at inference time.

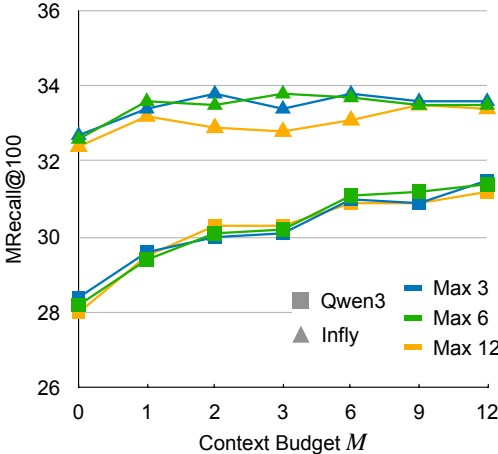

*Figure 5.* This figure shows the results of lowering verifier budget beyond 100 for Recall@100.

*Figure 6.* This figure shows the results of increasing the number of input documents on MRecall@100 for INF and Qwen3 models fine-tuned with different maximum document counts: 3, 6, and 12. We evaluate across 7 different context budgets.

### B.3. Extrinsic Verifier Evaluation

We further test RVR performance with an oracle verifier to determine which documents to use as context across $T = 5$ iterations. While the initial verified set $D_V$ is determined by our LLM verifier, context documents $D_{\text{ctx}}$ are selected by an oracle that ensures no two documents share the same answer string. Table **??** shows Recall@100 and MRecall@100 across five iterations. The results indicate that RVR performance can be further improved with better verification methods.

## C. Prompts & Trajectories

### C.1. Prompt for Qwen3 Verifier

The prompt we used to verify a document as gold using Qwen3-30B-Instruct is displayed in figure 8.

### C.2. Prompt for SearchR1

The prompt used for SearchR1 is displayed in Figure 9.

### C.3. Prompt for Tongyi

The prompt used for Tongyi is displayed in Figure 10.

### C.4. Trajectory for SearchR1

An example trajectory trace for SearchR1 is provided in Figure 11. For this specific example, the retrieval model is initial INF retriever $f_i$.

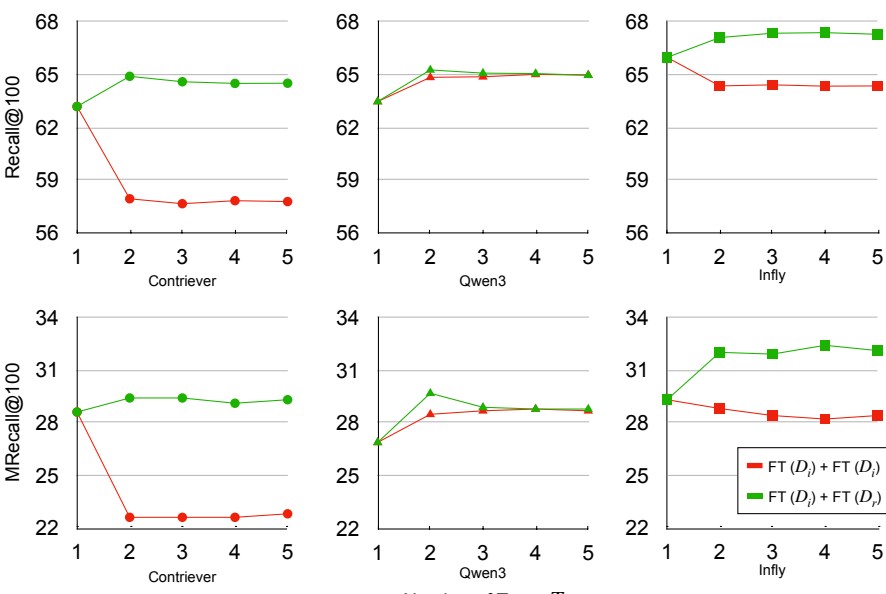

*Figure 7.* Multi-turn Generalization Results. This figure illustrates the change in Recall@100 and MRecall@100 across five iterations with a verifier budget of 50.

*Table 10.* Contribution analysis of first-stage and second-stage retrieval on QAMPARI test set. We report the average number of gold documents retrieved and the average number of new answers covered per question in each stage with verifier budget of 50.

| | | # New Gold Docs. | | # New Uniq. Ans. | |
|---|---|---|---|---|---|
| | | Turn 1 | Turn 2 | Turn 1 | Turn 2 |
| Contr-iever | FT $(D_i)$ + FT $(D_i)$ | 27.93 | 10.69 | 5.68 | 0.51 |
| | FT $(D_i)$ + FT $(D_r)$ | 27.93 | **13.98** | 5.68 | **1.05** |
| Qwen3 | FT $(D_i)$ + FT $(D_i)$ | 28.95 | 13.53 | 5.84 | 0.89 |
| | FT $(D_i)$ + FT $(D_r)$ | 28.95 | **13.61** | 5.84 | **0.93** |
| INF | FT $(D_i)$ + FT $(D_i)$ | 30.98 | 13.50 | 6.13 | 0.66 |
| | FT $(D_i)$ + FT $(D_r)$ | 30.98 | **15.09** | 6.13 | **1.04** |

*Table 11.* Comparing the performance of using oracle verifier vs. LLM verifier on QAMPARI test set. We evaluate MRecall@100 in RVR setting FT $(D_i)$ + FT $(D_r)$ with a verifier budget of 50.

| VERIFIER | CONTRIEVER | QWEN3 | INF |
|---|---|---|---|
| ORACLE | **29.60** | **30.50** | **33.50** |
| LLM (QWEN3-30B) | 29.40 | 29.70 | 32.00 |
| TOPK | 26.90 | 27.50 | 28.50 |

*Table 12.* Experimental Results on QUEST test set (N=1727). We report MRecall@100 (MR) and Recall@100 (R) for different retrieval models with a verifier budget of 50.

| | Contriever | | Qwen3 | | INF | |
|---|---|---|---|---|---|---|
| | MR | R | MR | R | MR | R |
| Base | 3.24 | 23.79 | 3.13 | 21.38 | 4.75 | 26.60 |
| FT ($D_i$) | 3.13 | 18.67 | 2.43 | 18.38 | 4.75 | 26.31 |
| **Ours: Retrieve-Verify-Retrieve** | | | | | | |
| Base + Base | 2.72 | 21.98 | 2.95 | 20.21 | 4.11 | 24.47 |
| Base + FT ($D_r$) | **3.82** | **22.89** | **4.05** | **23.88** | **5.73** | **28.71** |

## C.5. Trajectory for Tongyi

A portion of the trajectory trace for Tongyi is provided in Figure 12. Same as above, the retrieval model is initial INF retriever $f_i$.

---

**LLM Verifier Prompt**

**Prompt:** You are a document relevance verifier. Your task is to determine if a given document contains a direct answer to a specific question.

**Question**: *Harmony Korine was both screenwriter and director of what movie?*

**Document Title**: *Harmony Korine*

**Document Text**: *That Mr. Korine appears to be having it both (or many) ways may seem like a cop-out, but only if you believe that the role of the artist is to be a didact or a scold\", wrote \"The New York Times\". Principal filming wrapped up on March 30, 2012. The film was selected to compete for the Golden Lion at the 69th Venice International Film Festival. \"Spring Breakers\" received its world premiere at the 2012 Venice International Film Festival, and later was shown at the Toronto International Film Festival before being released to the general public in March 2013.*

**Instructions**:
- Answer with ONLY "YES" if the document directly answers the question or contains clear, specific information that answers the question.

- Answer with ONLY "NO" if the document does not contain a direct answer, is only tangentially related, or does not provide the specific information asked for

- Be strict in your evaluation. The document should contain factual information that directly addresses the query

Your answer (YES or NO):

*Figure 8*

---

**SearchR1 Prompt**

Answer the given question. You must conduct reasoning inside <think> and </think> first every time you get new information. After reasoning, if you find you lack some knowledge, you can call a search engine by <search> query </search> and it will return the top searched results between <information> and </information>. You can search as many times as you want. If you find no further external knowledge needed, you can directly provide the answer inside <answer> and </answer>, without detailed illustrations. For example, <answer> Beijing </answer>. Question: {question}

*Figure 9*

**Tongyi Prompt**

You are an expert in researching and answering complex questions. Your core function is to conduct thorough, multi-source investigations into any topic. You must handle both broad, open-domain inquiries and queries within specialized academic fields. For every request, synthesize information from credible, diverse sources to deliver a comprehensive, accurate, and objective response. Each of the questions has multiple answers. When you have gathered sufficient information and are ready to provide the definitive response, you must enclose the entire final answer within <answer></answer> tags. The answer should be a list of answers, separated by semi-colons.
# Tools
You may call the search tool one or multiple times to assist with the user query.
You are provided with the search tool, its signature within <tools></tools> XML tags:
<tools>
{"type": "function", "function": {"name": "search", "description": "Perform Google web searches then returns a string of the top search results. Accepts a single query.", "parameters": {"type": "object", "properties": {"query": {"type": "string", "description": "The search query."}}, "required": ["query"]}}}
</tools>
For each search tool call, return a json object with function name and arguments within <tool_call></tool_call> XML tags:
<tool_call>
{"name": search, "arguments": <args-json-object>}
</tool_call>

*Figure 10*

880
881
882 SearchR1 Example Trajectory
883
884 <|im_start|>system
885 You are a helpful assistant.<|im_end|>
886 <|im_start|>user
887 Answer the given question.     You must conduct reasoning inside <think> and </think> first every time you get new
information.      After reasoning, if you find you lack some knowledge, you can call a search engine by <search> query </
888 search> and it will return the top searched results between <information> and </information>.      You can search as many times
889 as your want.      If you find no further external knowledge needed, you can directly provide the answer inside <answer> and </
890 answer>, without detailed illustrations. For example, <answer> Beijing </answer>. Question: What manga was drawn by Ryoichi
891 Ikegami?
892 <|im_end|>
893 <|im_start|>assistant
<think>I need to find the manga drawn by Ryoichi Ikegami. I'll search for it.</think>
894 <search>Ryoichi Ikegami drawn manga</search>
895 <information>Doc 1(Title: "Ryoichi Ikegami") Ryoichi Ikegami is a Japanese manga artist. After graduating from junior high
896 school he moved to Osaka and drew manga while working as a billboard sign painter debuting at the age of 17 writing rental
897 comics. In 1966 he published a story called in the gekiga magazine Garo that caught the eye of fellow Garo contributor, manga
898 artist Shigeru Mizuki, who offered him a job as his assistant. Ikegami accepted and moved to Tokyo and worked as Mizuki's
899 assistant for two and a half years. In 2001, he won the Shogakukan Manga Award for general manga as the artist of "Heat".
Doc 2(Title: "Mikyo") Mikyo Mikyo
900 Doc 3(Title: "Manga iconography") Manga iconography Japanese manga has developed its own visual language or iconography
901 for expressing emotion and other internal character states. This drawing style has also migrated into anime, as many manga stories
902 are adapted into television shows and films. While this article addresses styles from both types of output, the emphasis here is on
903 the manga origins for these styles. The popular and recognizable style of manga is very distinctive. Emphasis is often placed on
904 line over form, and the storytelling and panel placement differ from those in American comics.
905 </information>
<think>I found out that Ryoichi Ikegami is a Japanese manga artist. Now I need to find which manga he drew.</think>
906 <search>Manga drawn by Ryoichi Ikegami</search>
907 <information>Doc 1(Title: "Manga") Manga Manga (Japanese: 漫画) are comics or graphic novel s originating from Japan. Most
908 manga conform to a style developed in Japan in the late 19th century, and the form has a long prehistory in earlier Japanese art.
909 The term "manga" is used in Japan to refer to both comics and cartooning. Outside of Japan, the word is typically used to refer to
910 comics originally published in the country. In Japan, people of all ages read manga.
911 Doc 2(Title: "Ryoichi Ikegami") Ryoichi Ikegami is a Japanese manga artist. After graduating from junior high school he moved
to Osaka and drew manga while working as a billboard sign painter debuting at the age of 17 writing rental comics. In 1966 he
912 published a story called in the gekiga magazine Garo that caught the eye of fellow Garo contributor, manga artist Shigeru Mizuki,
913 who offered him a job as his assistant. Ikegami accepted and moved to Tokyo and worked as Mizuki's assistant for two and a half
914 years. In 2001, he won the Shogakukan Manga Award for general manga as the artist of "Heat".
Doc 3(Title: "Manga iconography") Manga iconography Japanese manga has developed its own visual language or iconography
915 for expressing emotion and other internal character states. This drawing style has also migrated into anime, as many manga stories
916 are adapted into television shows and films. While this article addresses styles from both types of output, the emphasis here is on
917 the manga origins for these styles. The popular and recognizable style of manga is very distinctive. Emphasis is often placed on
918 line over form, and the storytelling and panel placement differ from those in American comics.
919 </information>
<think>I found out that Ryoichi Ikegami is a Japanese manga artist. Now I need to find which manga he drew. I'll search for his
920 notable works.</think>
921 <search>Ryoichi Ikegami notable works</search>
922 <information>Doc 1(Title: "Mikyo") Mikyo Mikyo
923 Doc 2(Title: "Ryoichi Ikegami") Ryoichi Ikegami is a Japanese manga artist. After graduating from junior high school he moved
924 to Osaka and drew manga while working as a billboard sign painter debuting at the age of 17 writing rental comics. In 1966 he
925 published a story called in the gekiga magazine Garo that caught the eye of fellow Garo contributor, manga artist Shigeru Mizuki,
926 who offered him a job as his assistant. Ikegami accepted and moved to Tokyo and worked as Mizuki's assistant for two and a half
927 years. In 2001, he won the Shogakukan Manga Award for general manga as the artist of "Heat".
Doc 3(Title: "Baiko") Baiko baiko
928 </information>
929 <think>I found out that Ryoichi Ikegami drew the manga "Heat". Now I can provide the answer.</think>
930 <answer> Heat </answer>
931
932 *Figure 11*
933
934

**Tongyi Example Trajectory**

```
{
  "type": "reasoning",
  "tool_name": null,
  "arguments": null,
  "output": "The API expects a single query string, not an array. So we need to provide a single query; perhaps we can combine
multiple queries separated by newline or something? However specification says: \"Perform a search for the top 10 results for
each query in one call.\" That suggests it can handle array of query strings. But error says 'query' must be a string. Perhaps the
wrapper expects a string named 'query', but earlier spec says Array of query strings. Let's try sending a single query that includes
multiple terms separated by semicolons.\n\nWe'll do one query: \"Ryoichi Ikegami manga list\". Let's try."
},

{
  "type": "tool_call",
  "tool_name": "search",
  "arguments": "{\"query\": \"Ryoichi Ikegami manga list\"}",
  "output": "A search for 'Ryoichi Ikegami manga list' found 5 results:\n\n## Web Results\n1. <docid: 1494481__1> [Ryoichi
Ikegami He became a professor at Osaka Uni...]\nRyoichi Ikegami He became a professor at Osaka University of Arts in 2005.
Ikegami has worked on several popular series, such as \"Mai, the Psychic Girl\" with writer Kazuya Kudo, \"Crying Freeman\",
with writer Kazuo Koike, as well as \"Sanctuary\" and \"Heat\" with writer Sho Fumimura. He also wrote and drew \"\", a manga
version of Spider-Man and collaborated with Garon Tsuchiya for the manga \"BOX\" (BOX 暗い箱). His most recent work is
\"Begin\" currently serialized in Big Comic Superior.\n\n2. <docid: 1494481__0> [Ryoichi Ikegami is a Japanese manga artist.
After ...]\nRyoichi Ikegami is a Japanese manga artist. After graduating from junior high school he moved to Osaka and drew
manga while working as a billboard sign painter debuting at the age of 17 writing rental comics. In 1966 he published a story
called in the gekiga magazine Garo that caught the eye of fellow Garo contributor, manga artist Shigeru Mizuki, who offered him
a job as his assistant. Ikegami accepted and moved to Tokyo and worked as Mizuki's assistant for two and a half years. In 2001,
he won the Shogakukan Manga Award for general manga as the artist of \"Heat\".\n\n3. <docid: 5676940__0> [Ryoko Yamagishi
is a Japanese manga artist. She is...]\nRyoko Yamagishi is a Japanese manga artist. She is one of the Year 24 Group, a collection
of female artists who innovated (girls') manga throughout the 1970s. Her major works include and \"Terpsichora\", both of which
have won manga awards. Biography. Ryoko Yamagishi was born on September 24, 1947, in Kamisunagawa, Hokkaido, Japan. As
a child, she studied ballet, which plays a part in many of her works. When she read the manga of Machiko Satonaka in 1964, she
decided to pursue becoming a manga artist.\n\n4. <docid: 66090580__0> [Ryū to Ichigo is a Japanese shogi -themed manga se...]
\nRyū to Ichigo is a Japanese shogi -themed manga series written and illustrated by Mitsuharu Yanamoto. It has been serialized in
Shogakukan's \"Weekly Shōnen Sunday\" since May 2020. Publication. \"Ryū to Ichigo\" is written and illustrated by Mitsuharu
Yanamoto. The series began in Shogakukan's \"Weekly Shōnen Sunday\" on May 20, 2020. Shogakukan has collected is chapters
into individual \"tankōbon\" volumes. The first volume was released on August 18, 2020, with a promotional video posted on the
same day. As of June 17, 2021, four volumes have been released.\n\n5. <docid: 65637267__0> [Ryoko (manga) Ryoko (stylized
as RYOKO) is a Japan...]\nRyoko (manga) Ryoko (stylized as RYOKO) is a Japanese manga series written and illustrated by
Kaito Mitsuhashi. It was serialized in Shogakukan's \"Weekly Shōnen Sunday\" from October 2016 to June 2019, with its chapters
collected in five \"tankōbon\" volumes. Publication. \"Ryoko\" is written and illustrated by Kaito Mitsuhashi. Mitsuhashi, at 21,
won the Shin Shedai Sunday Award (New Generation Sunday Award) for his work \"Ryoko Meet Meal!!\" in December 2015."
},
{
  "type": "reasoning",
  "tool_name": null,
  "arguments": null,
  "output": "The search result #1 is likely a summary of his works: \"Mai, the Psychic Girl\", \"Crying Freeman\", \"Sanctuary\",
\"Heat\", manga version of Spider-Man, \"BOX\", \"Begin\". Also maybe \"Kaidan Restaurant\"? Not sure.\n\nBetter to open
result #1."
},
```

*Figure 12*

