# OpenReview forum: "Retrieve-Verify-Retrieve: Comprehensive Document Retrieval via Training Retriever for Iterative Search"
_ICML.cc/2026/Conference — Submitted to ICML 2026_

### Official Review · Reviewer_8jZS · 2026-03-08

**Soundness:** 3
**Presentation:** 3
**Significance:** 2
**Originality:** 2
**Overall Recommendation:** 3
**Confidence:** 5

**Summary:**

This paper introduces Retrieve-Verify-Retrieve (RVR), an iterative retrieval framework that improves comprehensive document coverage for multi-answer queries. It works by verifying retrieved documents with an LLM and feeding them as context into a subsequent retriever specifically trained to find complementary, previously uncovered answers. RVR consistently outperforms both fine-tuned and agentic baselines by at least 10% relative improvement while being significantly more efficient. The main bottleneck lies in verifier quality, as performance plateaus after the second iteration, pointing to a clear direction for future work.

**Compliance With Llm Reviewing Policy:**

Affirmed.

**Final Justification:**

I would like to thank the authors for the rebuttal and extra experiments, which are really helpful for me.

My biggest concern is still the practical value of this work. As I mentioned and the authors reply in the first round of rebuttal, the simple ranking can achieve quite close performances with the proposed method with similar costs. It is applied to a similar case to RRF, where a simple RRF can achieve comparable performance to RVR (67.89 over 67.28, R@100 with INF).

The authors argue that their proposed method is complementary to these methods. However, these methods are simple, easy to implement, training-free, and can be generalized to other cases. I noted that the reviewer used Qwen-30B as the reranker, where I feel there might be a larger improvement when using larger/better LLMs for reranking compared to the same model applied to RVR as a verifier.

I feel the work itself is inspiring. However, given this concern, I would maintain my original judgement. I encourage the authors to go deeper to the design, which I feel has great potential.

**Key Questions For Authors:**

Please refer to the weakness points above. And one more question:
- How about using smaller K for Recall@k, e.g., QUEST also uses K=20, 50?

**Limitations:**

yes, there is the discussion of potential risks in the Impact Statement

**Strengths And Weaknesses:**

## Strengths
I appreciate the work from the following three perspectives:
- This paper proposes an effective iterative retrieval framework with a novel training objective. RVR trains the retriever to natively condition on already-retrieved documents and seek missing answers.
- RVR shows better performance (Table 1) with much lower cost (Table 2).
- RVR shows strong generalization (Table 4).

## Weaknesses
- (Major) The lack of baselines:
  - The idea of the authors uses the retriever to retrieve a set of documents first, and then further retrieve other documents based on the current retrieval results and the query. The core motivation is to improve the diversity of the retrieval. How about using a classical diversity-based retrieval method, i.e., Maximal Marginal Relevance (MMR) [1], which puts an extra term to encourage retrieval diversity during retrieval. I put the formula for the reference,

\begin{equation}
\text{MMR} = \arg\max_{d_i \in C \setminus S}
\left[ \lambda \cdot \text{sim}(d_i, q) -
(1-\lambda) \cdot \max_{d_j \in S} \text{sim}(d_i, d_j) \right]
\end{equation}

where $S$ is the set of already-selected documents, $C$ is the candidate
set, $q$ is the query, and $\lambda$ balances relevance and diversity.

  - If still following the idea of RVR, how about using instruction-aware retrieval, such as giving the prompt as part of the query input? My example for Figure 1 is below.
```
Retrieve documents that answer the query,
 but do NOT retrieve documents about
 Zack Snyder or Ariel Schulman,
 as these answers are already covered.
```
  - If we jump out of RVR and think of its fundamental mechanism - a combination between retrievers and LLMs, how about the performance of retrieve 2K and rerank with LLMs and keep top-K? The cost of this simple and straightforward method will be comparable with the cost of RVR (though more LLM calls, cheaper retrieval and index, plus training-free)

- (Major) The comparison with agentic baselines is unfair. SearchR1 and Tongyi are trained on single-answer multi-hop tasks (NQ, HotpotQA), while RVR is trained directly on QAMPARI for multi-answer coverage. The underperformance of agentic methods may reflect training distribution mismatch rather than a fundamental limitation of the agentic approach. A fair comparison would require retraining SearchR1 with a multi-answer coverage reward on QAMPARI.

- (Relatively minor) Still, other baselines will also be helpful to enhance the work:
  - Multi-Query Embedding, as mentioned in the literature
  - Query decomposition/expansion
  - Ensemble of different retrievers by the method, such as RRF - this is quite key to answer the question: whether the gain is from conditional retrieval or just one more time retrieval.

Overall, I appreciate the idea of iterative retrieval conditioned on LLM-verified documents, which is well-motivated and shows strong empirical results. However, the lack of important baselines and fair comparisons leaves key questions unanswered, and more experiments are needed to better justify the core contributions.

[1] The Use of MMR, Diversity-Based Reranking for Reordering Documents and Producing Summaries, SIGIR 1998

---

> ### Author Rebuttal · Authors · 2026-03-31
>
> ## W1: Comparison with MMR
> We experiment with S=200 candidates and various λ. We find that MMR does not improve performance (decreasing relevance (lambda) for diversity) monotonically hurts performance. Full results: https://imgur.com/a/YBOCqEi.
>
> ## W2: Instruction-aware retrieval
> We provide results with the suggested variation of RVR, extracting answer strings from verified documents with Qwen3-30B. We then retrieve a second round of docs using the prompt below, with {answer 1} to {answer n} replaced by the answers extracted.
>
> "Retrieve documents that answer the query, but do NOT retrieve documents about {answer 1} or {answer n}, as these answers are already covered. Query: {question}"
>
> We find this slightly underperforms our setting (conditioning on full document context) when using in-domain fine-tuned retrievers (FT(D_i)) for Qwen and Infly, slightly better for Contriever:
> |  | Contriever | | Qwen3 | | INF | |
>   |---|---|---|---|---|---|---|
>   |  | MR@100 | R@100 | MR@100 | R@100 | MR@100 | R@100 |
>   | Instruction Aware FT(D_i) + FT(D_i) | **29.50** | **63.79** | 27.80 | 64.24 | 30.30 | 66.16 |
>   | RVR FT(D_i) + FT(D_i) | 28.80 | 63.59 | **30.30** | **66.80** | **31.10** | **66.76** |
>
> ## W3: Retrieve 2K and rerank with LLM
> We retrieve 200 documents per query and prompt Qwen3-30B to rerank the documents by asking it to select documents that are relevant. We keep K=100 documents and discard the rest.
>
>  |  | Contriever | | Qwen3 | | INF | |
>   |---|---|---|---|---|---|---|
>   |  | MR@100 | R@100 | MR@100 | R@100 | MR@100 | R@100 |
>   | Reranked from FT(D_i) | **32.10** | 66.55 | 29.70  | 66.00 | 32.10 | 67.70 |
>   | RVR FT(D_i) + FT(D_r) | 31.60 | **66.83** | **31.40** | **67.28** | **33.70** | **68.70** |
>
> RVR outperforms this method for Qwen3 and INF and MR degrades slightly for Contriever. LLM reranking of 200 documents per query incurs a similar cost (0.94 s/q) to RVR's verification overhead (0.98 s/q).
>
> ## W4: Fairness of agentic baseline
> We agree with the reviewer and retrain SearchR1 on the QAMPARI dataset, setting the reward to be retrieving any correct answer:
> |  | Training data | Contriever | | Qwen3 | | INF | |
>   |---|---|---|---|---|---|---|---|
>   |  |  | MR@100 | R@100 | MR@100 | R@100 | MR@100 | R@100 |
>   | SearchR1 | Single-answer QA | 9.60 | 40.83 | 24.30 | 57.82 | 27.00 | 60.96 |
>   | SearchR1 | QAMPARI | 6.10 | 38.70 | 24.50 | 59.45 | 30.00 | 64.21 |
>   | RVR | QAMPARI | **31.60** | **66.83** | **31.40** | **67.28** | **33.70** | **68.70** |
>
> The in-domain agentic baseline improves over the original SearchR1 for Qwen3 and Infly, but drops for Contriever, likely due to it being a weaker model, and making it harder for the agent to learn to use it. This suggests the performance gap is not solely due to training distribution mismatch: iterative conditioning on verified documents provides a more effective mechanism for multi-answer coverage.
>
> ## W5a: Multi-query embedding
> We can use a multi-query embedding model (ColBERT) as base retriever. We present results below, with one modification: ColBERT does not allow long input query (out-of-memory issues), we use keywords from verified documents to expand the query rather than full document:
> |  | ColBERT |  |
>   |---|---|---|
>   |  | MR@100 | R@100 |
>   | Base | 33.30 | 68.85 |
>   | FT(D_i) | 37.60 | 70.85 |
>   | RVR Base + Base (keywords) | 37.40 | 72.01 |
>   | RVR FT(D_i) + FT(D_i) (keywords) | **38.50** | **72.43** |
>
> We observe that our RVR framework improves performance.
>
> ## W5b: Query decomposition/expansion
> Results using traditional query expansion:
> |  | Contriever | | Qwen3 | | INF | |
>   |---|---|---|---|---|---|---|
>   |  | MR@100 | R@100 | MR@100 | R@100 | MR@100 | R@100 |
>   | Query Expansion | 28.90 | 63.86 | 25.80 | 61.58 | 28.40 | 63.41 |
>   | RVR FT(D_i) + FT(D_r) | **31.60** | **66.83** | **31.40** | **67.28** | **33.70** | **68.70** |
>
> RVR outperforms query expansion.
>
> ## W5c: Retrieval fusion (RRF)
> Retrieval fusion is another approach that is complementary to ours. We provide RRF results (k=60):
>
> |  | MR@100 | R@100 |
>   |---|---|---|
>   | FT(D_i) (Contriever) | 28.60 | 63.19 |
>   | FT(D_i) (INF) | 29.30 | 65.99 |
>   | FT(D_i) w/ RRF | 32.80 | 67.89 |
>   | RVR (Contriever) | 31.60 | 66.83 |
>   | RVR (INF) | 33.70 | 68.70 |
>   | RVR w/ RRF | **34.30** | **69.56** |
>
> "FT(D_i) RRF" is where we fuse FT(D_i) results from Contriever (row 1) and INF (row 2). "RVR w/ RRF" is where we take "FT(D_i) w/ RRF" results, verify documents, and retrieve a second round conditioned on verified documents using FT(D_i) of Contriever and INF. Then we combine the two sets of second round results using RRF.
>
> ## Q1: Recall@K with smaller K
> We evaluate at K=20 and K=50. Results on QAMPARI with verifier budgets 100, 50, and 20 are here https://imgur.com/a/hp1Tskh. RVR outperforms baselines at all K values, with larger relative gains at smaller K.

---

> > ### Author Rebuttal · Reviewer_8jZS · 2026-04-02
> >
> > Thank you for the detailed rebuttal and for adding many new experiments. I appreciate the effort.
> >
> > I have one remaining question about practical value: based on the new results, a simple training-free retrieve-then-LLM-rerank pipeline seems broadly competitive with RVR at a similar cost (and in some settings, the gap appears small). Could you help clarify under what conditions RVR provides a clear and consistent advantage that justifies its additional training and pipeline complexity?
> >
> > This would help me better understand the practical trade-off and when RVR should be preferred over simpler alternatives.

---

> > > ### Author Response · Authors · 2026-04-06
> > >
> > > Thank you for the question. We agree reranking is a strong, simple baseline, and understanding when we would like to use RVR instead is important.
> > >
> > > 1. Orthogonality of gains: You can consider RVR as a general retrieval approach, and reranking can be applied to RVR outputs as well. Most retrieval systems improve with reranking. Just as with our experiments on ColBERT (multi query embedding) and RRF (mixing two retrievers), the gains from reranking are complementary to the gains from RVR. We provide additional experimental results below:
> > >
> > > | | Contriever | | Qwen3 | | INF | |
> > > |---|---|---|---|---|---|---|
> > > | | MR@100 | R@100 | MR@100 | R@100 | MR@100 | R@100 |
> > > | RVR w/ reranking (rerank 200 docs/round) | **33.10** | **68.31** | **32.10** | **68.02** | **34.90** | **69.84** |
> > >
> > > RVR with reranking improves over both standalone RVR and single-pass reranking alone. This mirrors the pattern we observed with RRF (W5c), where RVR w/ RRF outperformed both RVR alone and RRF alone. In both cases, the simpler alternative produces additional gains when combined with RVR, rather than replacing it.
> > >
> > > 2. Plateau in performance with reranking: We have tried reranking with a bigger reranking budget (up to 1K documents). Even with this higher budget, none of the base retrievers were achieving over 33.1 in MRecall. Given reranking relies on a single query embedding, we believe the reranking approach has a ceiling, especially when distinct target documents are far away from each other.
> > >
> > > 3. Different cost profile: We agree that the gains from reranking / RVR is comparable under the fixed latency. However, the cost profile of two approaches differ. Reranking requires more LLM costs, while RVR incurs fewer LLM calls but instead incurs additional retrieval calls. Therefore, in cases where additional search is cheaper than scaling LLM calls, RVR would be a more effective approach.
> > >
> > > 4. Choice of base model: We found that the choice of base retriever matters. Contriever is a relatively weaker model with limited query context window (N=512), thus the gains from RVR is somewhat limited.
> > >
> > > We will add these discussions in the updated manuscript. Thank you again for careful reading of our approach/paper and insightful questions!

---

### Official Review · Reviewer_xcsu · 2026-03-10

**Soundness:** 2
**Presentation:** 3
**Significance:** 3
**Originality:** 3
**Overall Recommendation:** 4
**Confidence:** 3

**Summary:**

This paper proposes the RVR (Retrieve-Verify-Retrieve) framework, an iterative document retrieval pipeline designed to solve the incomplete coverage problem in open-domain multi-answer queries.
RVR works in a loop: an initial retriever fetches candidate documents, an LLM verifier filters for relevant ones, verified documents are concatenated to the original query as context, and a specialized subsequent retriever (trained via contrastive learning to find missing content) retrieves supplementary documents.
Evaluated on QAMPARI and QUEST datasets against baselines like Tongyi DeepResearch and SearchR1, RVR achieves at least a 10% relative improvement and 3% absolute gain in complete recall, with only 2 retrieval calls per query (far more efficient than agentic methods with up to 16 calls).
It also demonstrates strong  out-of-domain generalization and includes comprehensive analyses of component roles, hyperparameter impacts, and verifier performance, revealing headroom for improvement with better verification models.

**Compliance With Llm Reviewing Policy:**

Affirmed.

**Final Justification:**

The authors have provided a thorough response and solid supplementary experiments. The section on statistical significance testing is especially well addressed. Accordingly, I have revised my score upward.

**Key Questions For Authors:**

1. Your evaluations are limited to Wikipedia-based text corpora for multi-answer QA. Do you anticipate the RVR framework to perform well in other retrieval scenarios (e.g., research paper retrieval, real-world web search), and what key adaptations would be required to the framework for these scenarios?

2. To enhance the credibility and reproducibility of the experimental results, could the authors provide quantitative statistical evidence to demonstrate that the advantages of the proposed method are statistically significant when compared with competing approaches?

**Limitations:**

The work has incremental novelty in LLM-retrieval integration: prior studies have already applied LLMs as retrieval result filters, and RVR’s key design of using verified documents for subsequent retrieval is a sequential extension of this existing approach, with the core innovative contribution in the retrieval-verification mechanism being evolutionary rather than a fundamental breakthrough.

**Strengths And Weaknesses:**

Strengths:
1. The framework exhibits superior retrieval effectiveness over all comparative baselines and boasts substantially higher efficiency in contrast to state-of-the-art agentic search approaches.These performance advantages are consistent across different base retriever models.
2. The paper carries out in-depth ablation studies, verifier performance evaluations, multi-round generalization tests, and hyperparameter analyses covering verifier budget and input context size. These comprehensive analyses not only clarify the role of each framework component and pinpoint key performance bottlenecks, but also offer actionable directions for subsequent optimization.
3. The framework demonstrates robust out-of-domain performance on the QUEST dataset even with base retrievers, mitigating the negative impacts of domain shift from in-domain fine-tuning, a critical merit for real-world deployment.

Weaknesses:
1. Evaluations are limited to only two Wikipedia-based multi-answer QA datasets (QAMPARI, QUEST) . There is no testing on other retrieval scenarios (e.g., research paper retrieval, real-world web search), limiting the generalization of the framework’s conclusions.
2. While more efficient than agentic search, RVR incurs additional overhead from verification and iterative retrieval (2-3x slower than one-shot baselines). And LLM-based verification in every iteration leads to notable time and token consumption, a key computational overhead that lacks in-depth analysis in the study.
3. One additional consideration for the work is the lack of statistical significance analysis accompanying the experimental findings. Though the RVR framework shows consistent performance improvements against baselines, the absence of relevant statistical metrics leaves an open question as to the statistical robustness of these observed results.

---

> ### Author Rebuttal · Authors · 2026-03-31
>
> ## W1/Q1: Limited to Wikipedia-based datasets; generalization to other scenarios
> We agree with the reviewer that evaluating on more diverse evaluation benchmarks would make the paper stronger. Unfortunately most retrieval benchmarks assume a single answer setting, making evaluation challenging.
>
> **We found another retrieval dataset, WebQuestionSP (N=1639)**, that assumes multiple answers, and reports results on this dataset below. While this still uses Wikipedia corpus, their query originated from the Google Suggest API with Freebase entities, a different query distribution than QAMPARI and QUEST. RVR shows consistent gains:
>
> |  | Contriever | | Qwen3 | | INF | |
>   |---|---|---|---|---|---|---|
>   |  | MR@100 | R@100 | MR@100 | R@100 | MR@100 | R@100 |
>   | Base | 62.16 | 77.38 | 61.00 | 76.06 | 62.47 | 77.39 |
>   | FT(D_i) | 49.45 | 65.72 | 46.68 | 61.50 | 51.60 | 67.06 |
>   | Tongyi | 52.33 | 65.84 | 57.56 | 71.80 | 58.29 | 72.37 |
>   | SearchR1 | 54.91 | 68.92 | 61.49 | 76.67 | 61.98 | 76.82 |
>  | Base + Base | **62.96†** | **77.93†** | **62.53†** | **77.54†** | **63.21†** | **78.20†** |
>   | Base + FT(D_r) | 61.49 | 76.81 | 60.81 | 76.28 | 62.72 | 77.91 |
>
> † = statistically significant over Base (we define our experiment for significance in **W3/Q2 Statistical Significance**).
>
> The inference setting mirrors QUEST, where FT(D_r) is trained on QAMPARI to test generalization. We note that Base + FT(D_r) does not improve over Base + Base on WebQSP, which we attribute to the domain mismatch.
>
> We acknowledge all three datasets use Wikipedia passage corpus. The RVR framework itself is corpus-agnostic: it requires only (1) a retriever that can search over a corpus and (2) a verifier that can judge document relevance. No architectural changes are needed for new domains, the key adaptation cost is fine-tuning the subsequent retriever on domain-specific training data.
>
> That said, to our knowledge there are no established multi-answer retrieval benchmarks for research paper retrieval or real-world web search that provide the gold answer string annotations required by our evaluation metrics (MRecall@K, Recall@K). Constructing such benchmarks and evaluating RVR on them is a valuable direction for future work.
>
> ## W2: Computational overhead from verification
>
> The reported overhead in our approach is primarily due to the verification process, as the reviewer correctly points out. Yet, we view the 2-3x overhead as a favorable tradeoff given consistent retrieval improvements, especially compared to agentic baselines that are 50-80x slower (Table 2).
>
> This overhead can be further reduced. The 2-3x overhead is primarily driven by verification. As verifiers examine individual documents respectively, we can parallelize the verification process. The input to each verification call is a single short document (~100 words) plus query, producing a binary label. We also use a relatively larger model (30B) as verifier, and one can use a smaller LM. We show the results of using a smaller LM (4B)  which performs similarly to the 30B model. Please see the response to Reviewer 2 (**W3/Q3: Verifier precision-recall tradeoff**).
>
> With B=100, the verifier examines 100 docs per round, but this can be reduced. Figure 3 shows that lowering B to 50 retains most performance gains across all retrievers, roughly halving verification time.
>
> ## W3/Q2: Statistical significance
>
> Thank you for the suggestion. We have conducted statistical significance testing using bootstrap resampling (10,000 trials, α = 0.05) across all main results. We provide the **results for QAMPARI** below.
> |  | Contriever | | Qwen3 | | INF | |
>   |---|---|---|---|---|---|---|
>   |  | MR@100 | R@100 | MR@100 | R@100 | MR@100 | R@100 |
>   | Base f_i | 19.00 | 54.17 | 16.70 | 52.94 | 26.10 | 62.34 |
>   | FT(D_i) | 28.60 | 63.19 | 26.90 | 63.48 | 29.30 | 65.99 |
>   | FT(D_i) + FT(D_i) | 28.80 | 63.59 | 30.30† | 66.80† | 31.10† | 66.76† |
>   | FT(D_i + D_r) + FT(D_i + D_r) | **31.70**†* | 66.12†* | 29.20† | 65.73† | 32.40† | 68.04† |
>   | FT(D_i) + FT(D_r) | 31.60†* | **66.83**†* | **31.40**†* | **67.28**†* | **33.70**†* | **68.70**†* |
>
>   † = statistically significant improvement over FT(D_i), * = statistically significant improvement over FT(D_i) + FT(D_i).
>
> The **results for QUEST** are below.
> |  | Contriever | | Qwen3 | | INF | |
>   |---|---|---|---|---|---|---|
>   |  | MR@100 | R@100 | MR@100 | R@100 | MR@100 | R@100 |
>   | Base | 3.24 | 23.79 | 3.13 | 21.38 | 4.75 | 26.60 |
>   | FT(D_i) | 3.13 | 18.67 | 2.43 | 18.38 | 4.75 | 26.31 |
>   | Tongyi | 0.93 | 10.51 | 2.49 | 19.52 | 3.30 | 21.95 |
>   | SearchR1 | 0.81 | 9.54 | 3.19 | 20.83 | 3.53 | 23.01 |
>   | Base + Base | 3.42 | 24.85† | 3.30 | 22.67† | 4.81 | 27.21† |
>   | Base + FT(D_r) | **4.52**†* | **26.01**†* | **4.40**†* | **25.43**†* | **6.02**†* | **30.53**†* |
>
>   † = statistically significant over Base, * = statistically significant over Base + Base.
>
> We conclude that RVR is statistically significant across our baselines.

---

> > ### Author Rebuttal · Reviewer_xcsu · 2026-04-03
> >
> > I appreciate the authors' detailed response and supplementary experiments. In particular, the statistical significance tests are well conducted and convincing. Based on the above, I have raised my evaluation score.

---

### Official Review · Reviewer_PMg7 · 2026-03-11

**Soundness:** 2
**Presentation:** 3
**Significance:** 2
**Originality:** 3
**Overall Recommendation:** 4
**Confidence:** 4

**Summary:**

This paper proposes Retrieve-Verify-Retrieve, a multi-round retrieval framework for comprehensive multi-answer retrieval. The core idea is to train a subsequent retriever (fr) that can explicitly retrieve answers not yet covered by previous rounds, using documents validated by LLM as conditions. Evaluations on in-domain dataset QAMPARI and out-of-domain dataset QUEST using three base retrievers show a 3-5 point improvement in MRecall@100 compared to the fine-tuned baseline, while with significantly lower computational cost than agentic baseline.

**Compliance With Llm Reviewing Policy:**

Affirmed.

**Final Justification:**

RVR proposes an iterative retrieval framework for multi-answer retrieval scenarios. My main concern was whether it can truly serve as a general iterative framework, since in the original experiments its performance appeared to saturate after T=2. The authors addressed this point directly in their discussion by conducting additional experiments with stronger verifiers like Qwen3.5-122B and GPT-5 nano. The results show that performance improves significantly at T=3 and beyond. This suggests that verifier quality is the main bottleneck, rather than an inherent limitation of the framework itself. In addition, the rebuttal has satisfactorily resolved my concerns regarding dataset coverage and the robustness of verifier selection. Therefore, I have raised my score from 3 to 4.

**Key Questions For Authors:**

Q1. The paper only uses one in-domain dataset (QAMPARI) and one out-of-domain dataset (QUEST). The absolute values on QUEST are extremely low (the best MR@100 is about 6%), making it difficult to support generalizable conclusions. The paper cites FanOutQA and NERetrieve as representative benchmarks for comprehensive retrieval in the Related Works section, but they are not used in the experiments and there is no explanation for their use. Why not supplement the experiments on these datasets?

Q2. Figure 2 shows that the performance of the LLM verifier completely saturates after Turn 2, meaning that only two retrievals are actually effective. The paper claims to propose a general iterative framework, but its actual contribution is only a two-pass retrieval. Under the current verifier design, is there any mechanism that can generate effective gains in the third round and beyond? If not, should the method be more accurately positioned as a two-pass retrieval?

Q3. The paper chose Qwen3-30B because it had the highest recall (74.05%), but its precision was only 34.06%, the lowest of the three. High recall and low precision mean a large number of noisy documents enter the fr context, which may be the cause of iteration saturation. However, switching to a high-precision verifier (such as GPT-5-nano) might cause the iteration to fail earlier due to missing gold documents. The paper did not explore this trade-off at all, making the explanation that "verifier causes iteration saturation" lack experimental support. What is the impact of different precision-recall combinations of verifiers on the efficiency of multiple iterations?

And the submission contains a minor presentation issue: Appendix B.3 contains an unresolved reference ("Table ??") that needs to be corrected.

**Limitations:**

No. The paper does not include a dedicated Limitations section. The authors should discuss the following:
1. The verifier saturates after Turn 2, making the iterative framework effectively a two-pass method in practice;
2. The optimal configuration has substantial GPU memory requirements, limiting practical applicability;
3. The method is evaluated only on list-style or set-seeking questions with well-defined gold answer sets, so its applicability to other question types remains unclear.

**Strengths And Weaknesses:**

Strengths:

1. Comprehensive multi-answer retrieval (RVR) is an overlooked but crucial problem. The paper clearly defines the task, and the design of the MRecall and Recall metrics is reasonable, effectively distinguishing between the capabilities of "covering most answers" and "covering all answers."

2. Table 2 shows that RVR requires only two fixed retrieval calls, with a total time consumption only 2-3 times that of the baseline, while Tongyi requires 100 times and SearchR1 requires 10 times. It significantly outperforms the agentic baseline while offering better performance, making this the paper's cleanest contribution.

3. The oracle verifier experiment in Table 8 clearly demonstrates the upper bound and headroom of the method. The contribution analysis in Table 6 quantifies the actual contribution of each retrieval round, and Figure 2 shows the saturation phenomenon after multiple iterations. These analyses do not deliberately conceal weaknesses.

Weaknesses

1. The paper only uses two datasets, QAMPARI and QUEST. The problem types are limited, and the absolute performance on QUEST is very low, which approximately 6% MR@100, insufficient to demonstrate the generalization ability of the method. Related work cites FanOutQA and NERetrieve but does not use them in the experiments, nor does it explain why.

2. Figure 2 shows that the performance of the LLM verifier completely saturates after Turn 2. The system is essentially a two-pass retrieval rather than an iterative framework. The paper attributes this to the verifier selecting redundant documents but provides no solution.

3. The paper chooses Qwen3-30B because it has the highest recall (74.05%), but its precision is only 34.06%, the lowest of the three. Low precision is likely the direct cause of iterative saturation in W2, but the paper does not explore the impact of different precision-recall combinations, nor does it report the end-to-end performance of smaller verifiers. The necessity of Qwen3-30B (57GB VRAM) lacks support.

4. Table 6 shows that the second round adds only an average of 0.33–0.77 unique answers, while the optimal configuration requires two retrieval models and indexes, consuming nearly 300GB of GPU memory. The paper does not discuss this cost-benefit trade-off.

---

> ### Author Rebuttal · Authors · 2026-03-31
>
> ## W1/Q1: Limited datasets; why not FanOutQA and NERetrieve?
> FanoutQA requires getting multiple intermediate documents for producing the final answer. However, FanOutQA is a QA benchmark evaluated on generation quality (accuracy, ROUGE, BLEURT), with no established retrieval evaluation. NERetrieve uses entity type descriptions as queries rather than natural language questions with answer strings. Our metrics require answer strings to judge document coverage, making evaluation on these benchmarks non-trivial.
>
> We agree with the reviewer that evaluating on more benchmarks would be good. Unfortunately most retrieval benchmarks assume a single answer setting, making evaluation challenging.
> We found another retrieval dataset, WebQuestionSP (N=1639). For results, please see the response to Reviewer 3 (**W1/Q1: Limited to Wikipedia-based datasets; generalization to other scenarios**).
>
> RVR shows consistent gains and all improvements are statistically significant. We will add WebQuestionSP results in the revised version.
> QUEST's low absolute MR@100 (~4-6%) reflects task difficulty, not an issue with our method. QUEST queries involve set operations with on average 10.5 answers. Base retrievers score 3-5%. RVR still achieves significant relative improvements ( 4.75 → 6.02 MR@100 with INF, 27% relative gain) and R@100 improves substantially (26.60 → 30.53).
> ## W2/Q2: Performance saturates after Turn 2; two-pass vs. iterative
> T=2 is optimal on QAMPARI with the current verifier, reflecting current verifier quality on this dataset, not a framework limitation. Figure 2 shows the oracle verifier continues improving through T=5 across all retrievers, identifying verifier quality as the bottleneck.
>
> Our framework accommodates mitigation strategies: verifier budget B and context budget M can be tuned per-round (Figures 3, 4), and any stronger verifier can be substituted without modifying retrieval components. We believe RVR remains a general iterative framework, oracle results demonstrate clear potential for additional rounds with stronger verification.
> ## W3/Q3: Verifier precision-recall tradeoff
> We thank the reviewer for the suggestion: systematically evaluating different verifiers would strengthen the paper. In the submitted paper, we didn’t experiment end-to-end results with alternative verifiers, as there were many design choices we were ablating.
>
> We make an assumption that a high recall verifier can be more beneficial than high precision verifier, as verifier serves two purposes: finding documents to contribute to the final output and identifying context for subsequent retrieval. High recall matters for both, missing gold documents means fewer correct outputs and less informative context. Low precision adds noise but does not remove gold documents.
>
> We run additional experiments using Qwen3-4B (higher precision / lower recall) instead of Qwen3-30B (lower precision / higher recall) verifier for one retrieval setting (INF):
> |  | **Verifier Qwen4B** | | **Verifier Qwen30B** | |
> |---|---|---|---|---|
> | | MR@100 | R@100 | MR@100 | R@100 |
> | FT D_i, T=2 | 34.20 | 68.81 | 33.70 | 68.70 |
> | FT D_i, T=3 | 35.00 | 69.08 | 34.20 | 69.09 |
>
> This alternative verifier performs similarly (we have conducted statistical significance testing using bootstrap resampling with 10,000 trials, α = 0.05, and found the difference between two variants are not significant). Using this verifier will also improve our efficiency computations (as it is substantially smaller). This indicates that our RVR framework is robust to the choice of verifier. Future work could conduct more rigorous experiments on finding the ideal verifier, studying the trade-off between precision, recall, and efficiency.
>
> Regarding VRAM: the verifier runs inference only on short inputs. With f_i = f_r, RVR uses about 131GB which is comparable to agentic baselines (~135GB).
> ## W4: Cost-benefit tradeoff
> The 300GB configuration is not required, it corresponds to the most expensive setting with INF and separate f_i, f_r models. Using a single retriever with RVR for both rounds (f_i = f_r) with Contriever requires 130GB with a single index, comparable to agentic baselines (135GB). We provide updated results of this experiment to show that it is comparable to using separate f_i,f_r models:
> |  | **Contriever** | | **Qwen3** | | **INF** | |
> |---|---|---|---|---|---|---|
> | | MR@100 | R@100 | MR@100 | R@100 | MR@100 | R@100 |
> | FT(D_i) + FT(D_i) | 28.80 | 63.59 | 30.30 | 66.80 | 31.10 | 66.76 |
> | FT(D_i + D_r) + FT(D_i + D_r) | 31.70 | 66.12 | 29.20 | 65.73 | 32.40 | 68.04 |
> | FT(D_i) + FT(D_r) | 31.60 | 66.83 | 31.40 | 67.28 | 33.70 | 68.70 |
>
> The single-model configuration (row 2) requires one index and one retriever while contributing up to 0.75 unique answers per query, comparable to the dual-model setup (rows 1 and 3, 0.33-0.77 unique answers). This demonstrates a favorable cost-benefit tradeoff: most of the performance gain can be achieved at roughly half the memory cost.

---

> > ### Author Rebuttal · Reviewer_PMg7 · 2026-04-04
> >
> > Thank you for the detailed rebuttal. The concerns in Q1 and Q3 are largely addressed. The rationale for excluding FanOutQA and NERetrieve is reasonable, and the additional results with the Qwen3-4B verifier further support the robustness of the method to verifier choice.
> >
> > However, Q2 remains my main concern. The rebuttal confirms that T=2 is optimal under the current verifier, and while the oracle verifier analysis is insightful, it relies on gold labels unavailable at test time. Could the authors experiment with a stronger verifier or multiple sampling with the current verifier to test whether meaningful gains beyond Turn 2 can be achieved? This would more directly support the claim that verifier quality is the bottleneck rather than a fundamental limitation of the iterative framework. I will maintain my current score pending this clarification.

---

> > > ### Author Response · Authors · 2026-04-06
> > >
> > > Thank you for the follow-up question. We experiment with a stronger open-source verifier (Qwen3.5-122B) and a closed-source verifier (gpt5-nano) to test whether verifier quality is the bottleneck for multi-turn gains. Due to budget and cost of gpt5-nano we experiment with INF only.
> > >
> > > | | Qwen3-30B (original) | | | | | | Qwen3.5-122B | | | | | | gpt5-nano | |
> > > |---|---|---|---|---|---|---|---|---|---|---|---|---|---|---|
> > > | | Contriever | | Qwen3 | | INF | | Contriever | | Qwen3 | | INF | | INF | |
> > > | | MR@100 | R@100 | MR@100 | R@100 | MR@100 | R@100 | MR@100 | R@100 | MR@100 | R@100 | MR@100 | R@100 | MR@100 | R@100 |
> > > | T=1 | 28.60 | 63.19 | 26.90 | 63.48 | 29.30 | 65.99 | 28.60 | 63.19 | 26.90 | 63.48 | 29.30 | 65.99 | 29.30 | 65.99 |
> > > | T=2 | 31.60 | 66.83 | 31.40 | 67.28 | 33.70 | 68.70 | 31.60 | 66.97 | 32.50 | 68.50 | 34.60 | 69.97 | 35.40 | 70.97 |
> > > | T=3 | 31.70 | 67.52 | 31.20 | 67.62 | 34.20 | 69.09 | 31.90 | 67.43 | 33.10 | 68.98 | 35.10 | 70.68 | **36.10** | 71.18 |
> > > | T=4 | 31.70 | 67.46 | 31.50 | 67.44 | 34.00 | 69.19 | 32.10 | 67.70 | 33.20 | 69.06 | 35.10 | 70.77 | 35.90 | **71.22** |
> > > | T=5 | 31.40 | 67.47 | 31.60 | 67.65 | 34.00 | 69.31 | **32.10** | **67.73** | **33.30** | **69.06** | **35.50** | **70.85** | 35.70 | 71.14 |
> > >
> > > With the original Qwen3-30B verifier, performance largely plateaus after T=2 as discussed. However, with stronger verifiers, we observe two things: (1) overall performance increases, and (2) we see gains beyond T=2. Qwen3.5-122B yields meaningful gains at T=3 across retrievers and continued gains at T=5 whereas the original version stagnates. We note that gpt5-nano shows a slight decline at T=4 & T=5, but still displays notable improvements beyond T=2 as well as large improvements over the original verifier results.
> > >
> > > We also note that our current retrievers are fine-tuned for a two-round setting: the subsequent retriever is trained to condition on a single set of verified documents from one prior round (Section 3.3) for simplicity. In the current setting, at T=3+, the retriever has to generalize to distribution unseen during training. Despite this, we still observe gains at T=3+ with better verifiers. We believe we will see more gains for longer horizons if we train a retriever with documents found from later rounds (2+).
> > >
> > > These results support our claim that verifier quality is the bottleneck rather than a fundamental limitation of the iterative framework and that meaningful gains can be achieved beyond T = 2. As verification improves, additional turns yield additional gains, confirming that RVR's multi-turn design can be meaningfully exploited with stronger verifiers. Thank you again for the thoughtful feedback and follow up questions!

---

### Official Review · Reviewer_YT9e · 2026-03-25

**Soundness:** 4
**Presentation:** 3
**Significance:** 2
**Originality:** 2
**Overall Recommendation:** 3
**Confidence:** 4

**Summary:**

To collect comprehensive documents for answers, multi-round retrieval framework is needed. The proposed method RVR equips the iterative retrieval framework with a document quality verifier, and differentiate the initial retriever and following one in their finetuning datasets. Experiment results on QAMPARI test set show the proposed iterative retrieval framework’s retrieval performance and efficiency among state-of-the-art baselines such as Tongyi and SearchR1.

**Compliance With Llm Reviewing Policy:**

Affirmed.

**Key Questions For Authors:**

1.	The concatenation of the query and the retrieved documents is taken as the new query for the next round in the proposed method. This will lead to the topic drift from the original query. How does the proposed method alleviate this problem?
2.	Is it necessary to differentiate the initial retrieval and the subsequent retrieval models? For the proposed method, they are finetuned with different training sets in Page3, Section 3.3.
3.	According to Table 2, the proposed method RVR achieve better retrieval performance with less retrieval calls. The large retrieval call gap between RVR and baselines indicates there is no need for multi-round retrieval calls if comprehensive results are collected during initial retrieval. Is it necessary to perform iterative retrieval?

**Limitations:**

yes

**Strengths And Weaknesses:**

Strength:

1.	The iterative retrieval framework is equipped with a document quality verifier. This is to avoid accumulative errors in the early retrieval results.
2.	Extensive experiments are conducted to show the necessary of introducing the verifier to the iterative retrieval process in terms of both effectiveness and efficiency.

Weakness:

1.	RVR emphasizes the initial and subsequent retrieval models are different. The motivation is unclear. Meanwhile, no empirical evidences in the experiments support this design.
2.	The comparison between RVR and baselines on QAMPARI test set show that 2 retrieval calls are enough to find as many related documents as possible to answer these test questions. The proposed iterative retrieval framework has limited application scenarios.

---

> ### Author Rebuttal · Authors · 2026-03-31
>
> ## Q1: Topic drift from concatenating query and retrieved documents
>
> This is a reasonable concern, but we think many of our design choices (described below) alleviate this concern. Empirically, we also report performance gains with document-augmented queries.
> 1. Training objective. The subsequent retriever is trained on inputs of the form [q; D_ctx] where D_ctx contains gold documents, and the training target is complementary gold documents related to q but distinct from D_ctx. This teaches the model to use context documents as signals for what has already been covered.
> 2. Verification as a filter. Unlike approaches that pass all retrieved documents forward, RVR only concatenates documents that the verifier has identified as relevant to the original query. Irrelevant or tangential documents are discarded before they can influence subsequent rounds.
> 2. Bounded context budget. The hyperparameter M limits the number of verified documents appended to the query (M=3 for Contriever, M=6 for Qwen3/INF). We show that performance does not improve, and can slightly degrade, beyond M=6, suggesting that a small, curated context is more effective than a large one.
>
> ## W1/Q2: Different initial vs. subsequent model
> We do not argue that having separate models for the initial, subsequent stage is necessary. Initial and subsequent stages implements slightly different retrieval function: the initial retriever takes only a query q and aims to return documents based on relevancy to the query (like standard retrievers), while the subsequent retriever takes a query augmented with documents [q; D_ctx], and aims to return documents based on relevancy to the query and distinctiveness from augmented documents.
> We train/present a single model that can handle both retrieval functions, by merging training data for both functions (FT(D_i + D_r) + FT(D_i + D_r) in Table 1  in the main paper). We note an error with an instruction for only the Qwen3 models in the submitted version, and present newer results below.
>
> |  | Contriever | | Qwen3 | | INF | |
>   |---|---|---|---|---|---|---|
>   |  | MR@100 | R@100 | MR@100 | R@100 | MR@100 | R@100 |
>   | OLD FT(D_i + D_r) + FT(D_i + D_r) | 31.70 | 66.12 | 26.30 | 65.73 | 32.40 | 68.04 |
>   | OLD FT(D_i) + FT(D_r) | 31.60 | 66.83 | 31.10 | 67.12 | 33.70 | 68.70 |
>   | NEW FT(D_i + D_r) + FT(D_i + D_r) | 31.70 | 66.12 | 29.20 | 65.73 | 32.40 | 68.04 |
>   | NEW FT(D_i) + FT(D_r) | 31.60 | 66.83 | 31.40 | 67.28 | 33.70 | 68.70 |
>
> Using a single retriever in both rounds is comparable to, and sometimes outperforms, using a different initial and subsequent retriever.
>
> ## W2: No need for more than two retrieval calls
>
> The reviewer correctly points out that we see minimal empirical gains after the second round retrieval in our current results. However, we do not view this as a fundamental weakness.
> 1. Multi-round (beyond 2 rounds) is beneficial when paired with a more powerful verifier. Figure 2 compares LLM vs. oracle verifier across five iterations. With the oracle verifier, both R@100 and MR@100 continue improving through T=5 across all three base retrievers. This demonstrates that the iterative framework itself has capacity for continued gains and the bottleneck is the verifier's ability to identify documents with novel answers.
> 2. Even with two rounds retrieval, the gains are substantial and broadly applicable. A single additional retrieval round produces 22-26 new gold documents and up to 1 new unique answer per query (Table 8), with consistent improvements across three retrievers on QAMPARI and two out-of-domain datasets (Table 4). Multi-answer retrieval is relevant to many real-world applications including comprehensive question answering, evidence aggregation, and systematic review all being scenarios where missing even one relevant document can matter.
>
> We view the efficiency of achieving strong gains in just two rounds as a practical strength: RVR provides meaningful coverage improvements at only 2-3x the cost of single-pass retrieval (Table 2), compared to 50-80x for agentic baselines
>
> ## Q3: The need for iterative retrieval
>
> The number of retrieval calls gap between RVR and baselines exists precisely because RVR achieves better results with fewer calls because of iterative retrieval. The key question is whether a single retrieval call can match RVR's coverage. Table 1 shows it cannot: even the strongest single-pass fine-tuned retriever (FT(D_i) with INF) achieves 29.30 MRecall@100, while RVR reaches 33.70, a statistically significant improvement.
>
> The single-pass retriever already retrieves 100 documents in one call; the issue is not retrieving more documents, but retrieving complementary ones that cover missing answers. Table 8 confirms this: Turn 2 contributes 22-26 new gold documents that the initial retriever missed, covering up to 1 additional unique answer per query.

---

### Decision · Program_Chairs · 2026-04-30

**Decision:**

Reject

**Comment:**

The paper received borderline reviews.

After the rebuttal, reviews remained borderline, with two of the reviewers recommending acceptance and two rejection.

The main issue raised that was not satisfactorily responded through the rebuttal is the lack of comparison with baseline methods and the apparent limited practical value of the approach.

The paper is technically sound. As it stands carries some value, although it would benefit from one more iteration.